# Harnessing the Potential of Exosomes in Therapeutic Interventions for Brain Disorders

**DOI:** 10.3390/ijms26062491

**Published:** 2025-03-11

**Authors:** Lu Bai, Leijie Yu, Mengqiong Ran, Xing Zhong, Meng Sun, Minhao Xu, Yu Wang, Xinlei Yan, Robert J. Lee, Yaqin Tang, Jing Xie

**Affiliations:** 1School of Pharmacy and Bioengineering, Chongqing University of Technology, 69 Hongguang Road, Chongqing 400054, China; 2Center for Nanomedicine and Gene Therapy, Chongqing University of Technology, 69 Hongguang Road, Chongqing 400054, China

**Keywords:** blood–brain barrier, exosomes, brain disorders, drug delivery, biocompatibility

## Abstract

Exosomes, which are nano-sized natural vesicles secreted by cells, are crucial for intercellular communication and interactions, playing a significant role in various physiological and pathological processes. Their characteristics, such as low toxicity and immunogenicity, high biocompatibility, and remarkable drug delivery capabilities—particularly their capacity to traverse the blood–brain barrier—make exosomes highly promising vehicles for drug administration in the treatment of brain disorders. This review provides a comprehensive overview of exosome biogenesis and isolation techniques, strategies for the drug loading and functionalization of exosomes, and exosome-mediated blood–brain barrier penetration mechanisms, with a particular emphasis on recent advances in exosome-based drug delivery for brain disorders. Finally, we address the opportunities and challenges associated with utilizing exosomes as a drug delivery system for the brain, summarizing the barriers to clinical translation and proposing future research directions.

## 1. Introduction

In recent decades, neurodegenerative disorders, cerebral neoplasms, and other neurological diseases have imposed a significant economic burden on society [1,2]. Research shows that in 2021, around 3.4 billion people (95% uncertainty interval: 3.2 billion to 3.62 billion) suffered from neurological disorders, accounting for 43.1% (40.5% to 45.9%) of the global population, leading to 11.1 million (9.75 million to 13.8 million) deaths [3]. In 2017, the economic burden of Parkinson’s disease in the U.S. was estimated at USD 51.9 billion, with projections suggesting a substantial rise by 2037 [4]. Nevertheless, the availability of clinically applicable diagnostic and therapeutic approaches remains inadequate, and the efficient delivery of pharmacological agents to the brain has emerged as a critical obstacle hindering advancements in the treatment of these brain disorders [5,6,7,8]. The robust structural integrity of the blood–brain barrier (BBB) alongside its selective permeability mechanisms presents significant challenges in drug delivery systems [9,10,11,12]. Composed of vascular endothelial cells and interconnected tight junctions, along with a complete basement membrane, pericytes, and astrocytes, the BBB meticulously regulates molecular transport within the central nervous system. This regulation is crucial for maintaining homeostasis, safeguarding the neural milieu, and preventing the ingress of blood cells, plasma constituents, and potentially harmful exogenous substances into the cerebral environment [13,14]. Due to these physiological conditions, the majority of pharmacological interventions designed to modulate pathophysiological processes or address central nervous system (CNS) disorders, as well as other cerebral conditions, are unable to permeate the BBB via systemic circulation. This limitation effectively restricts their access to the affected brain regions, thereby impeding optimal drug delivery. Consequently, there is an urgent clinical necessity for the development of an innovative drug delivery system capable of circumventing the BBB to enhance therapeutic effectiveness [15,16].

The significance of nanoparticle delivery technologies and alternative therapeutic approaches is becoming increasingly evident in this scenario. Nevertheless, nanocarriers that are synthesized from exogenous materials tend to exhibit various drawbacks, including cytotoxicity, a brief biological half-life, susceptibility to capture and elimination by the mononuclear phagocyte system, and poor penetration of the BBB [5]. Consequently, utilizing delivery systems derived from biological endogenous sources has emerged as a highly promising solution [17,18,19,20]. In recent years, extracellular vesicles (EVs)—minute carriers featuring a tightly sealed phospholipid bilayer structure—have garnered significant interest from the scientific community [16,21]. These EVs, originating from natural biological processes, exhibit remarkable intercellular transport capabilities and can be secreted by nearly all cell types via paracrine or autocrine mechanisms [22]. Acting as “messengers” of cellular communication, they encapsulate and convey a diverse array of bioactive molecules, including proteins, nucleic acids, and lipids, thereby establishing a sophisticated and intricate network of cellular communication within the organism [23], much like what we have studied in the past [24,25]. EVs are categorized into three distinct types based on their size, functionality, and release mechanisms: exosomes (ranging from 30 to 150 nm in diameter), microvesicles (spanning 100 to 1000 nm in diameter), and apoptotic bodies (exceeding 1000 nm in diameter) [26]. Exosomes present several advantages over synthetic nanocarriers. Primarily, exosomes exhibit superior drug-carrying capacity, accommodating both hydrophilic and hydrophobic drugs. Compared to conventional nanodelivery systems, such as polymeric nanoparticles and liposomes, exosomes can extend the blood circulation half-life [27]. Secondly, in terms of targeting mechanisms, exosomes demonstrate higher targeting efficiency than liposomes, attributed to their unique membrane structure and surface proteins. These specific proteins act as “barcodes”, guiding their recognition and efficient uptake by specific tissues or cells [28]. Thirdly, exosomes circumvent the risk of genomic integration compared to viral vectors [29]. Furthermore, naturally secreted exosomes exhibit inherent biocompatibility [30], stability [31], ability to penetrate physiological barriers [32], and low immunogenicity [33]. In contrast, synthetic nanocarriers often struggle to fully replicate the complexity and functionality of natural biomolecules, leading to immune responses from the host. Nevertheless, compared to traditional nanodelivery carriers, exosomes are limited by their natural membrane structure, and significant technical bottlenecks remain in drug loading efficiency and scalable production [34]. Currently, with increasing research into exosomes, they have emerged as biomarkers for disease diagnosis [35,36]. Furthermore, by functionalizing the surface of exosomes, targeted drug delivery using these carriers has become feasible [37]. These advantageous characteristics position exosomes favorably for the diagnosis and treatment of brain disorders. This manuscript intends to provide a thorough examination of the most recent advancements in scientific research pertaining to exosomes, with a particular emphasis on their investigation and application as diagnostic biomarkers and therapeutic carriers for neurological disorders. Initially, we systematically delineated the biological origins of exosomes and the fundamental principles governing various isolation techniques, conducting an in-depth analysis of the distinctive advantages and potential shortcomings associated with each method. Furthermore, we meticulously summarize the latest developments in innovative exosome loading strategies as drug carriers, alongside their functionalization modifications, which not only enhance the precision of drug delivery, but also augment therapeutic efficacy. Subsequently, this paper delves into the intricate mechanisms by which exosomes traverse the blood–brain barrier, elucidating their prospective capacity for transport in the context of treating brain diseases. This article comprehensively reviews how exosomes are involved in and influence the pathogenesis and progression of brain disorders, showcasing practical applications and prospects in the treatment of various brain diseases. These results not only enhance our comprehension of the biological roles of exosomes, but also facilitate the creation of innovative diagnostic instruments and therapeutic approaches for brain disorders. Additionally, by integrating recent research outcomes, the article emphasizes the remarkable ability of exosomes to traverse the blood-brain barrier, showcasing their considerable promise as biomarkers and delivery mechanisms for brain-related ailments. The insights and prospects outlined will contribute to the progression of our knowledge regarding exosome-based carriers in the precise delivery of medications for the management of brain disorders (Figure 1). Furthermore, a systematic bibliometric analysis revealed that over the past five years, the field of exosome-mediated brain delivery has seen the publication of more than a thousand original research articles (Web of Science Core Collection), with a growing number of review articles (PubMed database). Notably, current research exhibits significant limitations: the majority of publications concentrate on basic research within single disease entities (e.g., Alzheimer’s disease or glioblastoma), with a paucity of focus and commentary on the clinical therapeutic application of exosomes for related brain disorders [34,38]. Therefore, this review aims to comprehensively summarize and elucidate the research progress of exosomes in the treatment of brain diseases, followed by a review of the current status of relevant clinical applications, and the proposition of future challenges.

## 2. The Biogenesis of Exosomes

The production and secretion of exosomes represent a continuous biological mechanism. This intricate process primarily entails the invagination of the plasma membrane and the subsequent formation of multivesicular bodies (MVBs) (Figure 2). Exosome biogenesis initiates with the development of early sorting endosomes (ESEs), which arise from the invagination of the plasma membrane. During this phase, extracellular constituents—including proteins, lipids, small molecules, and metabolites—are internalized alongside cell surface proteins [39]. These sorting endosomes then progress to become late sorting endosomes (LSEs). The invagination of the LSE membrane leads to the generation of multiple intraluminal vesicles (ILVs), facilitating the incorporation of cytoplasmic components into these vesicles [40]. ILVs are composed of nucleic acids, proteins, and lipids that originate from the parent cell. Additionally, the Golgi apparatus and endoplasmic reticulum play crucial roles in the formation of ESEs and the transport of cargo [41]. At this stage, LSEs undergo further transformation into MVBs, which can either fuse with lysosomes or autophagosomes for degradation or associate with the plasma membrane to release ILVs as exosomes [41,42]. Therefore, the secretion of exosomes can be augmented by impeding the fusion with lysosomes [43]. In the context of biogenesis, the endosomal sorting complex required for transport (ESCRT) is crucial. It orchestrates molecular interactions and promotes membrane modifications. This mechanism results in membrane invagination, which is essential for the development of ILVs. The endosomal sorting complex required for transport (ESCRT) consists of class E vacuolar protein sorting (Vps) proteins that form four complexes: ESCRT-0, I, II, and III. ESCRT-0 to II primarily sort cargo into microdomains on the endosomal membrane, which act as centers for assembling molecular mechanisms that enhance signal transduction. ESCRT-III facilitates the budding and scission of these domains, resulting in the creation of intraluminal vesicles (ILVs). Vps4 is a AAA ATPase complex closely linked to ESCRT. After membrane scission, Vps4 aids in disassembling ESCRT III and interacts with it to stabilize the neck of developing intraluminal vesicles (ILVs) during membrane remodeling [44]. Research indicates that the RAB31 protein can transport the epidermal growth factor receptor (EGFR) into MVBs, contributing to ILV formation. Furthermore, RAB31 can obstruct the fusion of MVBs with lysosomes, which is one of the mechanisms of independent ESCRT [45].

## 3. Methods for Isolating Exosomes

EVs are lipid bilayer structures derived from the plasma membrane that are secreted by cells. They play a vital role in facilitating intercellular communication and shaping the tumor microenvironment. However, exosomes are often mixed with various contaminants in complex biological fluids, so how to effectively isolate and enrich relatively pure exosomes has become a key to research [46,47]. This article analyzes seven widely used methods for exosome separation and enrichment that have been implemented in recent years. These techniques include conventional ultracentrifugation, density gradient centrifugation, polymer precipitation, molecular exclusion chromatography, ultrafiltration, immunoaffinity chromatography, and microfluidic approaches. Furthermore, emerging technologies such as commercial kits, rinse separation methodologies, and fluidic systems are being explored to enhance the efficiency and speed of EV separations [48,49] (Table 1).

### 3.1. Ultracentrifugation

Ultracentrifugation (UC) has become widely acknowledged as the benchmark technique for the isolation of exosomes [50], accounting for 45.7% of the methodologies employed for exosome separation [51,52]. This technique operates on the principle of particle sedimentation in a solution, applying varying centrifugal forces to segregate sample components based on their density, size, and morphology. The sample is first subjected to processing based on its specific characteristics, during which larger biological particles are removed via low-speed centrifugation (e.g., 300× *g*). This is succeeded by a series of sequential centrifugation steps, employing centrifugal forces that range from 2000× *g* to 100,000× *g*. These steps are designed to efficiently eliminate cellular components, debris, apoptotic vesicles, and various contaminants, ultimately facilitating the isolation and purification of exosomes. This methodology is not only straightforward but also allows for large-scale exosome preparation using existing liquid concentration technologies [53]. However, ultracentrifugation may result in vesicle disruption, thereby affecting membrane structure and its biological activity [54].

### 3.2. Density Gradient Ultracentrifugation

Density gradient ultracentrifugation represents an enhanced separation methodology grounded in ultracentrifugation, predominantly utilized for the extraction and purification of exosomes. This methodology centers on creating a density gradient by employing two or more separation media with differing densities, such as sucrose and iodixanol. The standard protocol initiates with the removal of larger impurities through low-speed centrifugation, subsequently introducing the sample at the top of the density gradient medium [53,55], followed by a prolonged ultracentrifugation process that typically lasts over 16 h [56]. This sequential operation facilitates the aggregation of distinct particles in precise locations throughout centrifugation, predicated on their density disparities and relative densities to the medium, thereby yielding separations with elevated purity [57,58]. Given that exosome densities generally span 1.1 to 1.2 g/mL, this technique facilitates the effective isolation of exosomes from other extracellular components [59]. Despite the clear advantages of density gradient centrifugation in achieving relatively pure exosome separation, the procedure is cumbersome and time-consuming. Additionally, it does not effectively distinguish between vesicles and exosomes with similar densities [60,61,62].

### 3.3. Ultrafiltration

Ultrafiltration (UF) is a separation method predicated on molecular dimensions, recognized as one of the most efficient and uncomplicated approaches for the isolation of exosomes [63]. This methodology employs filter membranes characterized by diverse pore sizes or molecular weight cut-offs (MWCO) to remove contaminants from the sample; impurities exceeding the MWCO are retained, whereas exosomes that are smaller than the MWCO are permitted to traverse the membrane [60]. Ultrafiltration can be applied using various driving forces, including pressure, centrifugation, and electrical charge. In contrast to conventional ultracentrifugation methods, ultrafiltration markedly decreases the time required for exosome isolation and does not necessitate costly specialized equipment [64], making it a highly advantageous alternative. However, this method is often constrained by the membrane’s lifespan and transmembrane pressure, making it challenging to preserve exosomes in their native state [65,66].

### 3.4. Size Exclusion Chromatography

Size exclusion chromatography (SEC) is a separation methodology that capitalizes on variations in molecular size and is extensively employed for the extraction and purification of exosomes. This technique facilitates separation by introducing the sample into a column filled with porous media (such as Sephadex, Sepharose, and Sephacryl) and leveraging the hydrodynamic characteristics of these particles. Unlike ultracentrifugation, SEC does not rely on strong centrifugal forces, but rather separates by gravity or low-speed centrifugation, which helps maintain the biological function and integrity of exosomes [67]. In recent years, the SEC has been recognized as the most effective method for isolating morphologically and functionally intact exosomes from plasma [68]. However, exosomes isolated by SEC columns may exhibit a wide size distribution, especially in the smaller size range, and contaminants similar in size to exosomes may be present.

### 3.5. Polymer Precipitation

Polymer precipitation (PP) is an exosome separation technique based on the principle of superhydrophilic polymers (e.g., polyethylene glycol, PEG), which has gained popularity in exosome extraction in recent years. The technique facilitates the aggregation of exosomes through low-speed centrifugation by attaching the polymer to the water molecules encasing the exosomes, thereby diminishing the solubility of the solute [69]. This polymer precipitation approach is straightforward, does not necessitate sophisticated apparatus, is appropriate for processing substantial sample volumes, and effectively preserves the biological functionality of exosomes [70]. However, the purity of exosomes isolated through polymer precipitation methods is relatively low, primarily due to the co-precipitation of soluble non-exosomal proteins, viral particles, and other contaminants [51]. This contamination may lead to inaccuracies in exosome quantification, thereby affecting subsequent functional analyses.

### 3.6. Immunoaffinity

Immunoaffinity chromatography (IAC) is a separation technique based on antigen–antibody-specific reactions, which is widely used for exosome extraction and purification. The method utilizes the highly specific binding between specific proteins on the surface of exosomes (e.g., tetraspanin and ESCRT complex-associated proteins) and the corresponding antibodies [71], and realizes the selective capture of exosomes by immobilizing the antibodies on magnetic beads, chromatographic substrates or microfluidic devices [72]. The immunoaffinity method can significantly improve the purity of exosome isolation, and the isolated exosome has high specificity and good biological activity, but its operation process is relatively time-consuming, and the specificity and quality of the antibody are highly required. And the commonly used antibodies are mostly non-specific, which may lead to non-specific adsorption of exosomes [73]. The above reasons limit its application in large-scale studies.

### 3.7. Microfluidics

Microfluidics is considered to be a promising approach to enable miniaturization, integration, and high-throughput capabilities by integrating processes such as sample handling, analysis, and detection into a chip [74]. The technology enables the label-free isolation of exosomes by combining their physical and chemical properties (e.g., surface antigen, density, and size) with common separation methods. In addition, microfluidic devices can utilize external forces such as acoustic, magnetic or electric fields to further improve separation efficiency, and are becoming highly efficient platforms. In microfluidics, separation strategies based on physical properties have become a powerful approach. For example, exosome total isolation chip (ExoTIC) and deterministic lateral displacement (DLD) techniques have been developed for high-throughput exosome separation [75]. Furthermore, immunoaffinity microfluidic devices allow for the selective isolation of specific exosomes via antigen–antibody interactions, a technique that emphasizes the importance of choosing suitable antibodies to enhance microscale mass transfer and improve particle-surface interactions [76,77]. Despite the advantages of microfluidics, such as low sample consumption and short separation time, it still faces challenges such as sample clogging, low output, and high equipment cost.

A comprehensive review of current methodologies for exosome isolation and purification is lacking in the systematic inclusion of novel, engineered separation strategies [78]. One such technology, a novel high-throughput, label-free inertial microfluidic device (ExoArc), has been reported for the isolation of platelet-free plasma from blood for RNA and EV analysis. ExoArc has also been coupled with SEC to isolate EVs within 50 min, with a yield 10-fold higher than ultracentrifugation [78]. Furthermore, recent publications have described the use of an iodixanol–sucrose composite gradient system for density gradient centrifugation to isolate exosomes [79,80].

**Table 1 ijms-26-02491-t001:** Comparison of EV enrichment, separation, and purification methods.

Method	Principle	Advantage	Drawback	Purity	Time	Refs.
Ultracentrifugation	Components exhibiting disparities in size and density demonstrate a range of sedimentation velocities.	Gold standard; suiable for large-volume samples; ralatively cheap; mature.	Time-consuming; cumbersome operation; low yield; may damage exosomes.	Medium (with the co-precipitation and non-exosome contaminants).	>4 h.	[47]
Density gradient ultra-centrifugation	Based on EV density.	Good to maintain the activity of EVs.	Complexity; time-consuming.	High.	>16 h.	[81]
Ultrafiltration	The relative division employing various interception methods utilizes a sub-mass ultrafiltration membrane for the targeted separation of samples.	Low cost; fast speed; portable.	Membrane blockage and exosome loss; exosomes can be deformed or damaged.	High.	≈2–4 h.	[82]
Size exclusion chromatography	Separates exosomes based on hydrodynamic radii.	Preserve the integrity and natural biological activity; economical; good reproducibility.	Special columns and packing are required; lipoprotein.	High.	0.3 h.	[83]
Polymer precipitation	Highly hydrophilic water-separating polymers can alter the solubility of exosomes.	The polymer precipitation method is simple to operate; no need for complex equipment; suitable for handling large sample volumes; preserve the integrity and natural biological activity.	Lead to the wrong quantification; additional steps for higher; purity.	Low.	0.5–12 h.	[70,84]
Immunoaffinity	Based on antigen–antibody-specific recognition and binding.	Highly specific; easy to use; no contamination.	Low efficiency; Not suitable from large quantities; Expensive Nonspecific binding	High.	2–6 h.	[70,73]
Microfluidics	Based on different principles, including immunoaffinity, size, and density.	Highly efficient low cost; portable; easy to automate and integrate with diagnostics.	Low sample volume.	High.	-	[85]

## 4. Functionalization of Exosomes

Exosomes demonstrate significant potential as carriers for brain drug delivery due to their outstanding biocompatibility and low immunogenicity. However, their targeting capabilities remain inadequate [86]. To enhance the efficiency of exosomes in treating brain disorders, it is essential to functionalize them utilizing biotechnological approaches. Methods including genetic engineering, chemical techniques, and membrane fusion can improve targeting, optimize biodistribution, and increase blood–brain barrier (BBB) penetration.

### 4.1. Gene Engineering

Genetic engineering technology is a complex technology at the molecular level. To enhance the targeting capabilities of exosomes, researchers recombine genes for various targeting peptides in vitro and transfer them to recipient cells, enabling the secretion of exosomes with distinct targeting ligands. Sarkar et al. selected bone marrow-derived mesenchymal stem (BM-MSCs) cells to transfect plasmids encoding CAP-Lamp2b fusion protein. Through genetic engineering modification, exosomes have the ability to effectively cross the BBB [87]. Liang et al. studied the fusion of Her2-targeted protein to the N-terminus of Lamp2 protein in the exosome membrane, and cloned it into pLVX-GFP-N1, and finally fused it into a composite THLG (target-Her2-LAMP2-GFP), and then transfected HEK293 T cells with a lentiviral vector encoding THLG to isolate enhanced exosomes. Moreover, engineered exosomes loaded with anticancer drug 5-fluorouracil (5-FU) and miR-21 inhibitor oligonucleotide (miR-21i) can effectively target and improve anti-tumor ability [88]. Another study designed a plasmid vector encoding tLyp1-lamp2b fusion protein. Notably, tLyp1 is a tumor-homing penetrating membrane peptide, which can target tumors and penetrate into tumors. The tLyp1-exosome produced by cell transfection can target siRNA to cancer tissues [89]. The overexpression of amyloid-β precursor protein (APP) in the brain of Alzheimer’s disease (AD) leads to the binding of APP to Fe65 protein, which, in turn, triggers the secretion of amyloid-β, leading to the pathogenesis of AD. Iyaswamy et al. transfected pCI-Fe65 plasmid into HT22 hippocampal neuron cells to obtain exosomes overexpressing Fe65 [90]. Furthermore, they employed an autophagy inducer, corynoxin-b, for targeted delivery to APP-overexpressing neurons in the brain. This approach demonstrated potential in alleviating symptoms associated with AD.

### 4.2. Chemical Modification

The surface of exosomes can be functionalized by chemical modification. Tian et al. investigated the attachment of c(RGDyK) peptide to the exosome surface using bioorthogonal click chemistry. In the transient middle cerebral artery occlusion (MCAO) model, the engineered c(RGDyK) exosomes effectively targeted ischemic brain lesions [91]. Similarly, Jia et al. explored the attachment of RGE-targeted peptides (RGERPPR) to exosome membranes via click chemistry [92]. Following the administration in glioma models, RGE-exosomes successfully crossed the blood–brain barrier and targeted glioma. Cui et al. connected mesenchymal stem cell (MSCs)-derived exosomes with a rabies virus glycoprotein (RVG) peptide using a DOPE-NHS linker. After intravenous injection, RVG-exosomes improved targeting to the cortex and hippocampus, significantly enhancing the learning capabilities of Alzheimer’s mice [93]. Additionally, another study utilized vesicular stomatitis virus glycoprotein (VSVG) pseudotyping to create exosomes capable of recognizing surface biomarkers on target cells, thereby boosting the molecular functionality and targeting precision of exosomes [94].

### 4.3. Membrane Fusion

The lipid bilayer membrane of exosomes can integrate with various membrane structures to enhance the functionality of drug-loading systems. Lin et al. developed hybrid exosomes by co-incubating exosomes with liposomes, subsequently loading them with CRISPR/Cas9 expression vectors to modulate targeted gene expression [95]. In parallel, Rayamajhi et al. investigated the encapsulation of doxorubicin within hybrid exosomes formed from exosomes and liposomes. Their experiments demonstrated that drug-loaded exosomes exhibited increased toxicity against cancer cells [96]. Furthermore, a virus-mimic fusion vesicle (Vir-FV) can merge with exosomes, effectively identifying exosome miRNAs for clinical diagnostics and prognosis [97]. Takeda et al. explored the fusion of anionic liposomes with exosomes to elucidate the molecular mechanisms governing the fusion of intracellular exosomes and endosomes [98]. Mukherjee et al. proposed utilizing polyethylene glycol-modified cationic liposomes to enhance the binding to extracellular vesicles, thereby boosting the delivery efficiency of cancer therapeutics [99]. Another investigation employed bispecific chimeric antigen receptor T (CAR-T) cell-derived exosomes that fused with lung-targeted liposomes, resulting in the creation of a hybrid nanovesicle termed Lip-CExo@PTX, designated for targeted immunotherapy in lung cancer [100]. Lv et al. examined the application of genetic engineering to produce exosomes that express CD47, subsequently fusing them with thermosensitive liposomes. CD47 modification facilitates cargo evasion from the mononuclear phagocytic system (MPS) clearance. Furthermore, the integration of hyperthermic intraperitoneal chemotherapy (HIPEC) post-drug loading significantly enhances accumulation at tumor sites while augmenting the anti-tumor efficacy [101].

## 5. Strategies for Drug Loading in Exosomes

The unique biological attributes of exosomes, including their low immunogenicity, ability to traverse biological barriers, and inherent transport capabilities, render them attractive candidates for drug delivery applications [40,102]. The critical step in utilizing exosomes as drug delivery vehicles is the efficient loading of therapeutic agents into these vesicles [103]. Several methods exist, each with its own set of advantages and disadvantages. Furthermore, the loading strategies for diverse therapeutic agents exhibit variability. We will now elucidate the prevailing methodologies for drug encapsulation within exosomes (Figure 3). This delineation is intended to provide a framework for the selection of appropriate exosome loading techniques in subsequent investigations.

### 5.1. Incubation

The incubation method serves as a prevalent approach for loading drugs into exosomes. This technique facilitates the direct mixing of isolated and purified exosomes with pharmaceuticals, allowing for incubation under specific conditions that enhance drug molecule integration into the lipid bilayer or the internal compartments of the exosomes. Alternatively, this method enables the incubation of drugs with exosome donor cells, resulting in the direct acquisition of exosomes that secrete drug-loaded particles [105,106,107]. Incubation serves as a prevalent method for drug loading within exosomes due to its straightforward process, cost-effectiveness, and minimal impact on exosomal integrity and biological activity. Faruque et al. successfully synthesized Exo-PTX by incubating paclitaxel (PTX) with exosomes derived from human pancreatic ductal carcinoma cells, which were modified using the functional ligand RGD (peptide composed of several repetitions of Arg-Gly-Asp) and magnetic nanoparticles [108]. They found that Exo-PTX demonstrated enhanced cytotoxicity and a significant targeted anti-tumor effect in comparison to the control group. Beyond small molecule therapeutics, researchers can also integrate small nucleic acids and peptide-based drugs into exosomes via incubation. However, the limitations in drug loading efficiency and incubation conditions often require combining this method with others, preventing its standalone use.

### 5.2. Electroporation

Electroporation is a technique that employs an electric field to create temporary permeability in the exosome membrane, facilitating the entry of drug molecules into the exosome [109]. This technique facilitates exogenous drug loading into exosomes. After the separation and purification of exosomes, the drug integrates into them. Electroporation offers notable advantages. It efficiently loads drug molecules into exosomes and supports the loading of larger molecules as well. Currently, several studies have shown the effectiveness of electroporation for incorporating various drug molecules into exosomes. For example, Kim et al. delivered antisense miRNA oligonucleotides of miR-21 (AMO-21) into exosomes via electroporation for glioblastoma therapy [110]. Concurrently, the researchers fused T7 and Lamp2b onto the exosome membrane to create T7 peptide-modified exosomes (T7-exos), enhancing targeting efficiency. The findings demonstrated that T7-exos achieved a significant level of brain targeting, successfully traversing the blood-brain barrier, binding to transferrin on tumor cells, and substantially reducing tumor size. Similarly, Wu et al. employed electroporation to encapsulate 5-aminolevulinic acid hexyl ester hydrochloride (HAL) within exosomes derived from M2 macrophages [111]. This approach demonstrates substantial anti-inflammatory capabilities and can efficiently address atherosclerosis. Electroporation serves as a versatile technique not only for nucleic acid and small-molecule therapeutics, but also for the loading of macromolecular drugs, including proteins, into exosomes.

### 5.3. Sonication

Ultrasound serves as a non-invasive technique. It generates mechanical effects that render the exosome membrane temporarily permeable, facilitating the entry of drug molecules [112]. This ultrasonic method boasts several advantages. It offers straightforward operation, minimizes thermal effects on samples, and enhances the efficiency of drug loading while enabling continuous release in a short duration. Lamichhane et al. successfully facilitated the incorporation of HER-2-siRNA into exosomes by subjecting them to 35 KHz for 30 s, effectively down-regulating HER2 mRNA and protein expression levels [113]. However, this method can adversely impact exosome structure, leading to changes in spherical morphology and hydrophobic drug loading efficiency, as well as potential aggregation of exosomes [114]. Ultrasound can weaken the structural integrity and functionality of exosomes. Thus, it is crucial to optimize ultrasonic parameters—frequency, power, and processing time—to reduce damage and enhance drug loading efficiency. Additionally, combining the ultrasonic method with techniques like centrifugation may be necessary to preserve the purity and bioactivity of exosomes after drug loading.

### 5.4. Extrusion

Extrusion serves as a physical technique where a drug–exosome mixture traverses a lipid extruder fitted with a porous membrane of varying pore sizes (100–400 nm) under a regulated temperature. This process disrupts the exosome membrane, facilitating effective drug loading. It can regulate both exosome size and membrane consistency to a degree, achieving high drug loading efficiency [115]. The extrusion technique demonstrates a high efficiency for drug loading, yielding uniform-sized exosomes, making it a viable method for encapsulating drugs within exosomes [116]. Nevertheless, this process necessitates applying mechanical force to compress both the exosomes and the drug molecules for encapsulation. Such mechanical application may compromise the characteristics of the exosomal membrane, including its zeta potential and membrane protein architecture, thereby impacting the overall functionality of the exosomes [117]. Therefore, it is crucial to meticulously fine-tune parameters like pore size, extrusion cycles, and pressure during the extrusion process to minimize potential damage to exosomes while optimizing drug loading efficiency.

### 5.5. Transfection

Transfection involves employing a transfection agent to convey a specific plasmid into an exosome donor cell. This process facilitates the expression of the desired nucleic acid or polypeptide drug. Subsequently, the drug gets encapsulated within the exosome for secretion. Notably, this approach demonstrates superior drug loading efficiency and molecular stability compared to alternative methods [118]. Chen et al. transfected HeLa and HuH7 cell lines for 48 h utilizing human tumor virus-specific and hepatitis B virus-specific CRISPR/Cas9 expression plasmids, respectively [119]. They loaded Cas9 protein into exosomes, yielding successful acquisition of gRNA and Cas9 protein independently transported in endogenous exosomes. Additionally, the direct transfection of exosomes can yield exosomes containing drug molecules. For instance, de Abreu et al. demonstrated this by combining small extracellular vesicles (sEVs) with Exo-Fect, a commercial transfection kit, followed by incubation at 37 °C for ten minutes [120]. Their results revealed that Exo-Fect disrupts the sEV membrane, enhancing permeability. This process allows for miRNA loading into exosomes while also safeguarding miRNA from enzymatic degradation and improving intracellular transport and cargo delivery. Nonetheless, a significant drawback arises in the practical use of transfection—namely, the toxicity associated with transfection reagents. These agents may alter gene expression in exosomes derived from donor cells, consequently impacting the nucleic acid drugs and biological functions carried by exosomes. Thus, the quest for transfection reagents characterized by low toxicity and high biosafety remains imperative.

### 5.6. Freeze–Thaw Cycle

The freeze–thaw cycle technique serves as an advanced drug loading strategy that incorporates pharmaceuticals into exosomes. This approach entails a series of freeze-thaw iterations following the combination of exosomes and drugs. By exploiting temperature fluctuations, the technique enhances exosomal membrane permeability, facilitating the ingress of drug molecules [121]. The method involves cooling exosomes to low temperatures and subsequently thawing them at ambient conditions. This cycle prompts the exosomal membrane to rupture and undergo multiple repair phases. Throughout this process of repeated disruption and restoration, drug molecules seamlessly integrate into the exosomes, achieving effective loading. Significantly, the freeze–thaw cycle serves as a mild approach that maintains the membrane integrity of exosomes. Additionally, since exosomes exhibit stability at reduced temperatures, their intrinsic biological activity is preserved, making this method an attractive choice for large-scale production and enhancing its viability as a drug delivery mechanism [121]. However, it demonstrates lower drug loading efficiency when compared to ultrasound or extrusion methods. Additionally, exosomes treated via freeze–thaw cycles tend to exhibit larger particle sizes, likely due to aggregation during the process. Consequently, researchers often employ the freeze–thaw cycle method alongside other techniques to achieve improved drug loading outcomes.

### 5.7. Other Methods

Beyond the previously mentioned exosome delivery strategies, exosome drug loading can utilize methods such as thermal shock, low permeability dialysis, and chemical penetration. Thermal shock promotes drug molecule entry into exosomes through temperature alterations. This technique, also referred to as thermal shock treatment, employs elevated temperatures to modify exosome membrane fluidity, enhancing drug molecule membrane penetration [122]. The thermal shock method offers a significant advantage by enhancing drug loading efficiency within a short timeframe. Nevertheless, this approach necessitates the meticulous regulation of both temperature and duration to prevent adverse effects on the exosomal membrane’s integrity and functionality, as well as on the drug’s stability. Hypotonic dialysis involves using a dialysis membrane immersed in a hypotonic solution to encapsulate drug molecules within exosomes. The mechanism involves cell swelling induced by the hypotonic solution, which stretches the membrane, creates temporary pores, and boosts membrane permeability, thereby facilitating drug encapsulation within exosomes [123]. The chemical penetration approach involves utilizing agents like saponins to create pores in the exosomal membrane. This process facilitates the entry of drug molecules, enhancing the drug loading efficiency of exosomes [124]. Researchers have used various techniques to load drug molecules into exosomes, each with its own advantages and limitations (see Table 2 for the pros and cons of different exosome drug delivery technologies).

**Table 2 ijms-26-02491-t002:** Advantages and disadvantages of exosome drug loading strategies.

Loading Strategy	Pros	Cons	Refs.
Incubation	The procedure remains straightforward, incurs minimal expenses, and preserves the integrity and biological functionality of exosomes with minimal impact.	Inefficient and will retain unwanted content, subsequent need to purify the removal of unloaded drugs.	[104,125]
Electroporation	It has high loading efficiency and can load drugs with large molecular weight.	It will damage the structure and function of exosomes and cause exosome aggregation.	[109,126]
Sonication	Simple operation. Drug loading and continuous drug release are highly efficient.	It changes the structure of exosomes and causes exosome aggregation.	[114]
Extrusion	The drug loading efficiency is high, and the size of the drug-loaded exosomes is uniform.	Mechanical force may destroy the properties of exosome membrane and affect the integrity of exosome function.	[115]
Transfection	Compared with other methods, transfection makes drug loading efficiency and molecular stability higher.	Transfection agents have certain toxicity and safety problems, which may lead to changes in gene expression of donor cells that produce exosomes.	[118,120]
Freeze–thaw cycles	The process is mild, maintaining the integrity of the exosome membrane, and maintaining higher cell viability.	It is low in efficiency and easy to cause exosome aggregation.	[121,127]
Thermal shock	Load a large number of drugs in a short time	It has a negative impact on the integrity and function of exosome membrane and the stability of drugs.	[122]
Chemical penetration	High loading efficiency	It is difficult to completely remove chemical reagents, and may lead to exosome membrane permeability and increase cytotoxicity.	[123]

## 6. Mechanisms of Exosomal Penetration Through the BBB

The blood–brain barrier, recognized as a highly specific physiological protective structure, fundamentally relies on brain capillary endothelial cells [128]. These cells adeptly coordinate both passive diffusion and active transport mechanisms to ensure the efficient and selective delivery of hormones, ions, and essential nutrients into the brain. Concurrently, they meticulously regulate gaseous molecules such as oxygen and carbon dioxide within the brain and its blood circulation, thereby maintaining the homeostasis of the cerebral environment and supporting normal physiological functions [129,130].

Recent studies have demonstrated that exosomes can traverse the blood–brain barrier, facilitating transport from the bloodstream into the brain; however, the precise mechanisms behind this penetration remain incompletely understood. Substantial evidence indicates that exosomes primarily navigate the BBB via transcytosis, although some may utilize the paracellular route, allowing them to pass through brain cells via the extracellular space [131]. Chen et al. investigated the intricate mechanisms by which exosomes traverse the BBB [132]. Their findings reveal that cell-derived exosomes primarily navigate this barrier via brain microvascular endothelial cell (BMEC) endocytosis, leading to their accumulation within BMECs in the endosomal compartment. This process is notably enhanced under conditions resembling a stroke. The predominant outcome for these internalized exosomes is the formation of MVBs, which subsequently fuse with the plasma membrane to facilitate the release of cargo into the extracellular space, thereby ensuring effective delivery to target cells. Furthermore, they also noted that exosomes can be internalized through clathrin-mediated endocytosis, caveolae-mediated endocytosis, and macropinocytosis. Additionally, a minor fraction of exosomes manages to reach the brain via interendothelial cell spaces. Saint-Pol et al. elucidated five potential mechanisms governing the communication and interaction between exosomes and neural cells [133,134], and Figure 4 provides a detailed depiction of exosome penetration mechanisms and their subsequent fate pathways. Furthermore, the process of exosome traversal across the BBB, under physiological or pathological conditions, may be constrained by exosome size and specific surface ligand characteristics, a process intricately regulated by the BBB state and microenvironmental factors [135,136].

### 6.1. Disruption of the BBB Impacts Exosome Penetration Efficiency

Under pathological conditions such as brain injury, inflammation, and tumors, the integrity of the BBB is significantly compromised, providing exosomes with increased access to brain tissue [137,138]. Recently, the Kim team developed an ultrasound-responsive nanoparticle (BTNP-pDA-BNN6) that generates nitric oxide (NO) under high-intensity focused ultrasound. This NO production modulates matrix metalloproteinase-9 (MMP-9) expression, leading to ZO-1 damage and disruption of BBB integrity. Furthermore, they demonstrated that the BBB opening induced by BTNP–pDA–BNN6-mediated NO release is transient, with the barrier re-establishing within approximately 2 h [139]. Consequently, their research suggests the possibility of real-time, controllable, and selective “turning on” and “turning off” the BBB in specific brain regions. Kefeng Zhai et al. investigated the potential of ginsenoside Rg1 to ameliorate brain vascular endothelial damage, observing that during traumatic brain injury (TBI), macrophage-derived exosomes containing miR-21 downregulated the expression of ZO-1, occludin, and claudin-5 in HBMECs, disrupting tight junctions and consequently reducing BBB permeability. Their research also indicated that macrophage-derived exosomes containing miR-21 exhibited enhanced BBB transmigration in the TBI state [21]. Alessandro Villa et al. demonstrated that EVs derived from gliomas possess a remarkable capacity to penetrate the BBB and accumulate within brain tumors [140].

### 6.2. Exosome Trafficking in Brain Metastasis

Cancer cells generate a large number of exosomes that play a crucial role in cancer cell invasion and migration via angiogenesis. Emerging evidence indicates that exosomes derived from triple-negative breast cancer cells exhibit high metastatic potential in the brain due to the presence of the lncRNA gene *GS1-600G8.5*, which contributes to the disruption of the blood–brain barrier (BBB) and enhances its permeability [141]. Furthermore, studies by Golnaz Morad et al. have demonstrated that exosomes derived from breast cancer cells disrupt the integrity of the BBB through transcellular endocytosis. Further investigation suggests that this mechanism is dependent on a clathrin-mediated endocytic pathway [136].

### 6.3. Regulation of Transcytosis

The molecular mechanisms of transcytosis are influenced by a variety of regulatory factors. Banks et al. conducted an investigation into the efficiency of BBB transport utilizing 10 exosome samples derived from both murine and human sources [142]. They examined the influence of lipopolysaccharide (LPS), Wheatgerm lectin (WGA), and mannose 6-phosphate (M6P) on exosome traversing across the BBB. Their findings revealed that while all exosome variants displayed the ability to penetrate the BBB, the degree of penetration varied significantly, with differences observed up to 10-fold among various exosome types. The LPS can enhance the transcytosis of exosomes across the BBB through various mechanisms, including the stimulation of cytokine release and the disruption of prostaglandin-dependent pathways. WGA promotes the adsorption-mediated transcytosis of glycoproteins that possess sialic acid and N-acetyl-D-glucosamine, whereas compounds that interact with the M6P receptor may impede this transcytosis across the BBB. Rab11 recycling endosomes play a crucial role in the regulation of transcytosis, functioning to transport cargo carried by exosomes to the basolateral membrane and mediating subsequent exocytosis. Specifically, the depletion of Rab11 leads to the accumulation of recycling endosomes, including the accumulation of key molecules such as transferrin receptors [143,144].

## 7. Exosome-Based Therapies for Brain Disorders

### 7.1. Exosomes and Acute Neurodegenerative Disorders

Exosomes significantly influence intercellular communication within the CNS. They transport biological molecules, like proteins and nucleic acids. This transport is crucial for understanding brain disease mechanisms and identifying new diagnostic and treatment targets. Exosomes also serve vital physiological roles in immune regulation, tissue repair, regeneration, drug delivery, and act as biomarkers. Their complex functions in vivo shed light on their diagnostic and therapeutic potential. Notably, exosomes can cross the BBB, enhancing their applicability in treating various brain disorders. This document summarizes and discusses the role of exosomes in specific brain diseases. The goal is to establish a robust foundation for future research and clinical applications while providing valuable insights.

#### 7.1.1. TBI

Traumatic brain injury (TBI) is a kind of brain injury and neurological dysfunction caused by severe external impact. It is divided into primary injury and secondary injury [145]. Specifically, brain cells die rapidly after being severely impacted by external forces, which destroys the cerebrovascular system and leads to primary injury such as cerebral hemorrhage, which in turn leads to inflammation, oxidative stress and neuronal death [146]. Exosomes are involved in intercellular communication and play an important role in the recovery of nervous system injury. They are effective biomarkers for TBI [147]. Almost all cells in the brain can secrete exosomes, including microglia, neurons, astrocytes, mesenchymal stem cells, and plasma. Exosomes secreted by these cells are potential strategies for the treatment of TBI [148]. A study revealed that both in vitro TBI models and murine TBI models demonstrated that activated astrocyte exosomes facilitate the shift of microglia from the M1 pro-inflammatory state to the M2 anti-inflammatory state [149]. This process largely results from the accumulation of miRNA-873a-5p within exosomes, which suppresses the release of pro-inflammatory mediators from microglia by inhibiting the ERK and NF-kB phosphorylation pathways. Consequently, this action mitigates brain injury in TBI-affected mice and enhances neurological recovery post-injury. In both murine and neuronal TBI models, the administration of HucMSC-derived exosomes improved neurological function, reduced cerebral edema, and curtailed apoptosis and ferroptosis triggered by TBI [150]. Liu et al. proposed incorporating bone marrow mesenchymal stem cell-derived exosomes (BME) into hyaluronic acid hydrogel to simulate the release of brain matrix and exosomes. Their findings demonstrate that the hybrid hydrogel facilitates synergistic endogenous neural stem cell (NSC) recruitment and differentiation, alongside angiogenesis, within the injured regions of TBI rats, thereby promoting neurological functional recovery following TBI [151] (Figure 5). Another investigation revealed that human neural stem cells (hNSCs) and their exosomes significantly improved neurological behavior and elevated markers of nerve regeneration in TBI rat models [152]. Chen et al. utilized interferon-gamma (IFN-γ)-pretreated neural stem cells (NSCs) to produce derived exosomes (IFN-Exo). They integrated IFN-Exo into a collagen/chitosan scaffold via 3D printing technology, which preserved exosomal activity. Experimental results demonstrated that the exosomes mitigated neuroinflammation and enhanced neurological recovery after TBI through modulation of the MAPK/mTOR signaling pathway [153]. Additionally, Wang et al. established that plasma-derived exosomes possess potential therapeutic effects for TBI, with various microRNAs enriched within these exosomes potentially serving as novel biomarkers [154].

#### 7.1.2. Stroke

Stroke is a dangerous acute cerebrovascular disease with high mortality and disability. Its etiology includes both vascular occlusion and rupture. Clinically, it categorizes into ischemic and hemorrhagic strokes. An insufficient or halted cerebral blood supply results in neuronal damage, leading to the loss of brain functionality. Given the severe implications, neurovascular remodeling and neurological recovery post-stroke hold paramount importance. Exosomes, as essential mediators of intercellular signaling, play a pivotal role in the series of tissue responses that follow brain injury. They possess the ability to traverse the BBB, serving as vital biomarkers for stroke diagnosis, management, and prognosis [155]. Zhang et al. used neural stem cells (NSCs) and their derived exosomes to treat cerebral ischemia caused by middle cerebral artery occlusion in mice, which can better improve cerebral infarction and neuronal death, and contribute to the recovery of neurological function after stroke [156]. Research demonstrates that utilizing exosomes derived from human neural stem cells to encapsulate brain-derived neurotrophic factor (BDNF) effectively reduces oxidative stress in neural stem cells (NSCs). This process also fosters the differentiation of NSCs into neurons, aiding nerve recovery post-ischemic stroke [157]. Moreover, Wu et al. used astrocyte-derived exosomes (ATC-Exo) to transport miR-34 c and simulated cerebral ischemia/reperfusion (I/R) injury in N2a cells in vitro, and found that miR-34c transported by ATC-Exo alleviated I/R-induced neuronal injury [158]. Additionally, not only can cell-derived exosomes (CDE) serve as treatment options for stroke, but plant-derived exosome-like nanoparticles (ELN) also exhibit therapeutic potential and yield higher production rates. Specifically, Panax notoginseng saponins (PNSs) enhance recovery of neurological functions in patients suffering from acute ischemic stroke, improving their daily living activities [159]. Furthermore, Li et al. obtained exosome-like nanoparticles from Panax notoginseng, demonstrating their ability to traverse the BBB and modify microglial phenotypes, thereby reducing I/R injury in ischemic stroke therapy [160].

Ferroptosis plays an important role in neurodegenerative diseases such as stroke. A crucial gene in the ferroptosis pathway for ischemic stroke patients is CHAC1. Wang et al. developed anti-ferroptosis exosomes from adipose-derived mesenchymal stem cells (ADSC-Exo) [161]. These exosomes contain miR-760-3p, which effectively suppresses CHAC1 expression in neurons. By targeting CHAC1, it enhances the inhibition of ferroptosis, thereby improving neurobehavioral outcomes in mice post-ischemia/reperfusion (I/R) (Figure 6). Similarly, Yi et al. used miR-19b-3p to modify the exosomes of ADSCs, and found that they reduced ferroptosis and nerve injury in a mouse model of cerebral hemorrhage, which was related to miR-19b-3p and its targeting iron regulatory protein [162]. Additionally, scientists have engineered a highly effective M2 microglia-targeted exosome (M2pep-ADSC-Exo) that demonstrates significant targeting specificity for M2 microglia in both in vivo and in vitro studies. This exosome can further mitigate the ferroptosis of M2 microglia and safeguard the neurological function of mice suffering from ischemic stroke [162].

### 7.2. Exosomes and Chronic Neurodegenerative Diseases

Neurodegenerative diseases are a group of progressive neurological disorders with complex mechanisms driving their onset and progression. Key factors include protein misfolding and aggregation, oxidative stress, mitochondrial dysfunction, neurotransmitter imbalances (like dopamine and acetylcholine), and neuroinflammation. Abnormal protein accumulation is crucial in many neurological disorders. For example, Alzheimer’s disease features β-amyloid (Aβ) plaques and misfolded tau proteins, while misfolded tau is also found in Parkinson’s disease (PD) and Huntington’s disease (HD). In amyotrophic lateral sclerosis (ALS), TAR DNA-binding protein 43 (TDP-43) is involved (Figure 7). Research highlights the significant role of exosomes in the formation and transmission of these abnormal proteins, with studies showing their potential in diagnosing and treating neurodegenerative diseases.

#### 7.2.1. Alzheimer’s Disease

In the realm of neurological disorders, Alzheimer’s disease (AD) stands out as a prime subject of investigation in the field of exosome research. The hallmark of this disease involves the aberrant modification and aggregation of amyloid-beta (Aβ) and tau proteins within the brain [164]. Exosomes play a multifaceted role in the pathological progression of AD, including the propagation of Aβ and tau proteins [165,166], the modulation of neuroinflammation, and the facilitation of intercellular communication [166,167], thereby exacerbating the disease’s pathological cascade. Consequently, exosomes present themselves as potential therapeutic targets for novel AD interventions.

Nucleic acids and proteins within exosomes are considered sensitive biomarkers for AD diagnosis. Rastogi et al. observed that salivary exosomes from individuals with cognitive impairment exhibited significantly elevated levels of Aβ oligomers, monomers, and phosphorylated tau antibodies compared to healthy controls [168]. Jia et al. reported that neuronal exosomes from AD patients displayed markedly higher levels of Aβ42, total tau, and pT181-tau compared to other groups [169]. McKeever et al. demonstrated that the levels of miR-125b-5p, miR-451a, and miR-605-5p in cerebrospinal fluid exosomes from patients with early-stage dementia were significantly different from those in normal individuals [170]. These findings provide a crucial foundation for the early diagnosis of AD.

To date, no therapeutic interventions have demonstrated efficacy in preventing or reversing the progression of AD. Research indicates that exosomes hold promise as drug delivery vehicles for Alzheimer’s disease therapeutics [90] (Figure 8). Studies by Jiang et al. have demonstrated the capacity of exosomes to serve as effective carriers for delivering drugs and nucleic acid fragments, such as siRNA and miRNA, across the BBB [171]. Alvarez-Erviti et al. showed that exosomes loaded with a β-secretase inhibitor (BACE1 siRNA) reduced Aβ production [172]. Losurdo et al. reported that the intranasal delivery of MSC-derived exosomes significantly reduced neuroinflammation and tau pathology, and improved memory function in 3xTg AD mouse models [173]. Yan et al. improved cognitive function in APP/PS1 mice by upregulating exosomal miR-451a and miR-19a-3p [174]. Furthermore, Šála et al. developed a neutral sphingomyelinase 2 inhibitor, PDDC, which inhibits exosome release and is associated with AD pathology [175]. Research has indicated that exosomes derived from hippocampal neural stem cells can protect hippocampal synapses from the toxicity of Aβ oligomers, restoring long-term potentiation (LTP) and memory function, thus providing a novel approach for AD treatment [176]. Chen et al. found that reducing the secretion of pathological exosomes significantly decreased Aβ production, thereby delaying disease progression [177]. These findings highlight the promising potential of exosome applications in AD therapy, offering new avenues for future development and research.

#### 7.2.2. Parkinson’s Disease

Parkinson’s disease (PD) is a prevalent neurodegenerative disorder of the central nervous system, characterized by the degeneration and loss of dopaminergic neurons and the formation of Lewy bodies [178]. Studies have revealed that the levels of exosomes and phosphorylated α-synuclein in the saliva of PD patients are significantly elevated compared to healthy controls [179]. Exosomes, acting as carriers for α-synuclein propagation, play a crucial role in PD pathogenesis. Research indicates that exosome-associated α-synuclein oligomers are more readily internalized by cells than their free counterparts and can induce neurotoxicity, leading to the death of dopaminergic neurons [180]. Furthermore, microglia-derived exosomes can induce motor deficits and neurodegeneration, and by activating microglia, they inhibit the clearance of α-synuclein, thereby promoting its accumulation and propagation [181]. Collectively, these findings suggest that exosomes not only enhance α-synuclein propagation but also promote microglial activation in the progression of PD [182,183].

The diagnosis of PD currently relies heavily on clinical presentation, with a paucity of effective auxiliary diagnostic tools. However, components of exosomes, including miRNAs, proteins, and lipids, may serve as biomarkers for PD. Research by Jiang and Chang demonstrated that neuronal exosomes isolated from serum effectively discriminate between PD and atypical parkinsonism. The α-synuclein content in exosomes increases progressively with PD progression, potentially enabling its use in PD diagnosis or disease progression monitoring [184,185]. Studies by Wang et al. suggest that exosomal markers LRRK2 and Rab0 protein in urine and cerebrospinal fluid (CSF) may also function as PD biomarkers [186]. Zhao et al. found significantly elevated miR-331-5p and decreased miR-505 in plasma exosomes from PD patients [182]. Research by Caldi Gomes indicated that miR-126-5p, miR-99a-5p, and miR-501-3p in CSF could serve as PD markers [187].

The etiology of PD remains elusive, and current therapeutic interventions are limited in their ability to halt disease progression. Consequently, elucidating the underlying pathological mechanisms of PD and identifying novel therapeutic strategies are of paramount importance. Yang et al. administered exosomes loaded with the antisense oligonucleotide sequence ASO4 into the brain ventricles of PD mice, resulting in a significant reduction in α-synuclein aggregation, attenuated degeneration of dopaminergic neurons, and a subsequent improvement in motor function [187]. Liu et al. utilized a cellular model of α-synuclein-induced neuronal damage and demonstrated that α-synuclein nanoscavengers facilitated the degradation and clearance of intracellular α-synuclein via autophagy, thereby effectively preventing neuronal loss [188]. Chen et al. reported that exosomes secreted by human umbilical cord mesenchymal stem cells (hucMSCs) promoted the survival of dopaminergic neurons, reduced neuroinflammation, and improved motor function in a PD model through the induction of autophagy [189]. Ye et al. found that exosomes derived from umbilical cord blood significantly ameliorated PD-related pathological features by modulating the protein expression levels of the MAPK p38 and ERK1/2 signaling pathways [190]. Shukla et al. demonstrated that under PD-related stress conditions, exosomes derived from cells expressing hsa-miR-320a are actively internalized into recipient cells, a process that rescues cell death and mitochondrial reactive oxygen species (ROS) production in recipient neurons and glial cells [191]. Furthermore, exosomes derived from human neural stem cells effectively prevent 6-OHDA-induced toxicity in SH-SY5Y cells by reducing intracellular ROS and the expression of pro-inflammatory factors through the downregulation of related apoptotic pathways [192]. Esteves et al. found that exosomes derived from umbilical cord blood mononuclear cells, acting as biological carriers, delivered miR-124-3p, which protected dopaminergic neurons in the substantia nigra and striatal fibers, thereby significantly improving motor function in injured mice [193] (Figure 9). Kojima et al. reported on engineered exosomes that mitigated the neurotoxicity and neuroinflammation of Parkinson’s disease through the delivery of therapeutic catalase mRNA [194].

#### 7.2.3. Amyotrophic Lateral Sclerosis

Amyotrophic lateral sclerosis (ALS) is a chronic and progressive neurological disorder. It primarily affects upper and lower motor neurons, as well as the trunk, limbs, and facial muscles [195]. In recent years, studies have shown that the development of ALS may be mediated by changes in spinal cord intercellular communication between neurons and glial cells. One of the possible ways in which intercellular communication occurs is through exosomes [196]. Recently, Goldschmidt-Clermont et al. highlighted the significant therapeutic potential of exosome vesicles derived from Schwann cells in addressing ALS [197]. In their investigation, they utilized allogeneic Schwann cell-derived exosome vesicles for sustained intravenous infusion in patients with advanced ALS. The findings indicated that these exosome vesicles could effectively ameliorate motor neuron dysfunction associated with ALS. Initial evaluations have confirmed the safety of this treatment approach. Additionally, a considerable body of research has concentrated on the therapeutic impact of stem cell-derived exosomes on ALS. Bonafede et al. investigated the therapeutic potential of exosomes derived from adipose-derived stem cells (ASCs) in the treatment of ALS. Their findings demonstrated that these ASC-derived exosomes improved motor performance and reduced glial cell activation in SOD1 (G93A) mice. These results suggest the potential of ASC-derived exosomes for therapeutic applications in ALS [198]. In addition, Garbuzova-Davis and Borlongan et al. demonstrated that EVs derived from human bone marrow endothelial progenitor cells (hBMEPC) can alleviate plasma-induced injury in endothelial cells (EC) within ALS mouse models [199].

### 7.3. Neuroglioma

Glioma represents a primary brain tumor, thought to arise from neural stem cells or progenitor cells harboring tumor initiation genes [200]. Despite the use of surgical intervention, radiotherapy, and alkylating chemotherapy showing some efficacy, glioma prognosis remains bleak. This situation correlates with the intricate heterogeneity of the glioma tumor microenvironment (TME). The complex heterogeneity of glioma TME hinges on the communication network among glioma cells and adjacent stromal cells [201]. In this context, exosomes serve a significant function as ‘communication messengers’ between cells. Exosomes serve as vital mediators of intercellular communication. They assist in tissue repair, immune modulation, and the transfer of metabolic products to target cells. However, the dissemination of aberrant materials can trigger pathological conditions, including cancer, metabolic disorders, and neurodegenerative diseases [202]. Thus, exosomes hold significant importance in the progression, diagnosis, chemoresistance, and management of glioma. For instance, Xu et al. demonstrated that miR-3184-3p exhibits high levels in the cerebrospinal fluid exosomes of glioma patients and decreases following tumor excision [203]. MiR-3184 drives glioma advancement through two mechanisms. First, it directly facilitates cell proliferation, migration, and invasion while also suppressing apoptosis in glioma cells. Second, miR-3184 within glioma-derived exosomes shifts macrophages toward M2-like phenotypes. Furthermore, research conducted by Lu et al. revealed that exosome microRNA-671-3p enhances cell proliferation in glioma by directly targeting CKAP4 [204]. EMT significantly contributes to glioma progression. Research indicates the involvement of exosomes in glioma EMT. For instance, Chen et al. demonstrated that exosome-derived microRNA-708 enhances glioma cell proliferation and EMT by inhibiting the SPHK2/AKT/β-catenin pathway [205]. These findings underscore the pivotal role of exosomes in glioma development and proliferation.

In the conventional treatment of glioma, chemotherapy serves as a crucial method. Drug resistance significantly impacts its efficacy. Exosomes also play a vital role in this context. Zeng et al. report that exosomal miRNAs might contribute to the resistance of glioblastoma(GBM) cells to TMZ [206]. The researchers assessed the expression levels of miR-151a in two TMZ-resistant GBM cell lines. Their findings indicate that heightened resistance to TMZ correlates with reduced expression of miR-151a. Furthermore, the extent of chemotherapy resistance in GBM tumors links to the quantity of exosomes containing miR-151a present in cerebral spinal fluid. Yu et al. reported that the transfer of miR-199a to mesenchymal stem cells derived from glioma cells via exosomes can inhibit the proliferation, invasion, and migration of glioma cells [207]. Additionally, the researchers discovered that high expression of miR-199a in mesenchymal stem cells restored glioma cell sensitivity to TMZ. Furthermore, miR-199a inhibited glioma cell proliferation by down-regulating AGAP2, demonstrating a significant therapeutic effect.

The aforementioned studies indicate that exosomes significantly influence the onset, growth, diagnosis, and drug resistance of glioma. For instance, in 2018, Jia et al. successfully loaded superparamagnetic iron oxide nanoparticles (SPIONs) and curcumin (Cur) into exosomes. They then utilized click chemistry to link the exosome membrane with a neuropilin-1 targeting peptide, creating glioma-targeted exosomes capable of both imaging and therapeutic functions. Upon administering these exosomes to glioma cells and in situ glioma models, they noted that the modified exosomes effectively traversed the BBB. Moreover, these exosomes demonstrated promising outcomes for targeted imaging and treatment of GBM [92]. Katakowski et al. transfected the miR-146b plasmid into bone marrow stromal cells (MSCs) to isolate the exosomes released by MSCs. They subsequently injected these exosomes into tumors. The results indicated that this approach considerably reduced glioma development in primary brain tumors within the employed mouse model [208]. Lee et al. utilized exosomes isolated from U87MG human glioblastoma cells to encapsulate a novel anticancer agent, selumetinib, aimed at enhancing glioma treatment [209]. Their findings indicated that, in animal models of GBM, drugs delivered via GBM-derived exosomes were evaluated against both non-exosome-encapsulated counterparts and exosomes not specific to GBM, in both in vitro and in vivo settings. The surface characteristics of GBM-derived exosomes enabled them to transport therapeutics directly to GBM tumor sites, achieving cytotoxic effects without the need for additional targeting peptides, while remaining non-toxic to healthy brain cells. This underscores the unique ‘homing’ capability of exosomes as effective carriers for drug delivery.

In addition to directly loading pharmaceuticals into exosomes for glioma treatment, advancements in nanotechnology have encouraged researchers to explore co-loading with various nanocarriers. These carriers include liposomes and polymer nanoparticles. This approach enhances the diversity of drug loading and improves therapeutic efficacy. For instance, Zhang et al. employed endothelial cell-derived exosome-coated doxorubicin-loaded nanoparticles for the immunogenic chemotherapy of glioblastoma. This research examined the efficacy of exosome-coated doxorubicin (DOX) nanoparticles (ENPDOX) in terms of BBB permeability, induction of immunogenic cell death (ICD), and enhancement of survival in GBM mouse models. Findings indicated that mice receiving ENPDOX demonstrated dendritic cell maturation, cytotoxic cell activation, cytokine modulation, proliferation inhibition, and increased apoptosis of GBM cells in vivo. Consequently, ENPDOX prolonged survival in GBM-affected mice, suggesting its potential as a promising treatment strategy for glioblastoma [210]. Recently, Hao et al. utilized a hybrid nanovesicle combining NK cell-derived exosomes and RSL3-loaded liposomes to direct ferroptosis-immunotherapy, achieving a synergistic enhancement of anti-glioma efficacy [211]. In this research, they engineered a specific hybrid nanovesicle (hNRV) integrating NK cell-derived extracellular vesicles with RSL3-loaded liposomes aimed at glioma treatment (Figure 10). Thanks to the enhanced permeability and retention effects of NK exosomes and their tumor-targeting properties, the results indicated that hNRV actively accumulated in tumors and improved cellular uptake. This significantly facilitated the release of FASL, IFN-γ, and RSL3 into the tumor microenvironment, wherein FASL from NK cells effectively induced tumor cell lysis. RSL3 decreased GPX4 expression in tumors, resulting in the buildup of LPO and ROS, thereby inducing ferroptosis in malignancies. Furthermore, the heightened levels of IFN-γ and TNF-α promoted dendritic cell maturation, effectively led to GPX4 inactivation, encouraged lipid peroxidation, and increased sensitivity to ferroptosis, indirectly fostering ferroptosis occurrence. The interplay between these factors brought forth a beneficial cycle of ferroptosis and immunotherapy. Furthermore, Exosomes exhibit promising potential in preventing glioblastoma after surgery.

The aforementioned research indicates that exosomes significantly influence the onset, progression, diagnosis, drug resistance, and treatment of glioma. Regarding therapy, the role of exosomes extends beyond their intrinsic capabilities; they serve primarily as drug delivery vehicles. These vesicles can transport various therapeutic agents, including nucleic acid drugs, chemotherapy agents, and protein-based treatments, demonstrating considerable potential in glioma management [212,213]. In summary, as nanotechnology advances, exosomes emerge as a promising therapeutic avenue for glioma.

### 7.4. Exosomes and Other Brain Disorders

#### 7.4.1. Exosomes and Intra-Cranial Infections

Intra-cranial infections are inflammatory disorders of the central nervous system instigated by a diverse array of pathogenic microorganisms, including viruses, bacteria, fungi, spirochetes, parasites, rickettsiae, and prions [214]. These pathogens breach the blood–brain barrier, infiltrating the cerebral parenchyma and causing various inflammatory pathological manifestations such as encephalitis, meningitis, and brain abscesses [215].

Japanese encephalitis (JE) is an inflammation of the brain induced by the *Japanese encephalitis virus* (*JEV*), which has a mortality rate of 25–30% [216]. Extensive research has demonstrated that JEV is transmitted through mosquito bites, subsequently infiltrating human dermal layers and regional lymphatic tissues. Infected cells migrate from the periphery to the central nervous system, evading peripheral immune surveillance through multiple mechanisms, including the suppression of major histocompatibility complex (MHC) presentation and interference with interferon signaling pathways [217,218,219,220]. Regarding the mechanism by which *JEV* traverses the BBB to infiltrate the central nervous system, Adjanie Patabendige and colleagues employed a BBB model created using human brain endothelial cells (HBECs) and human astrocytes. Their findings indicate that infections, such as *JEV*, lead to an elevation in inflammatory mediators, thereby compromising BBB integrity and facilitating viral infiltration into the brain [221]. Several studies have noted that following *JEV* infection, activated microglia can induce neuronal apoptosis by releasing EVs, thereby contributing to the pathogenesis observed in the brain. In a therapeutic study [222], Bian et al. utilized mesenchymal stem cell (MSC) transplants in JEV-infected mice, which successfully decreased microglial activation, encouraged phenotypic switching, safeguarded BBB integrity, and impeded the progression of JE [223]. Given the propensity of MSC therapy to induce vascular obstruction, exosomes derived from MSCs, characterized by their smaller particle size and lower immunogenicity, exhibit significant potential for JE treatment.

*Tick-borne encephalitis virus* (*TBEV*), the causative agent of tick-borne encephalitis (TBE), is an RNA virus classified within the Flaviviridae family and is categorized as a flavivirus [224]. Based on variations in genetic sequences, antigenic properties, and geographical prevalence, TBEV is subdivided into several subtypes: European (*Eu-TBEV*), Far East (*FE-TBEV*), Siberian (*Sib-TBEV*), Himalayan (*HiM-TBEV*), and Baikalian (*Bkl-TBEV*) [225,226,227]. Clinically, TBEV infection presents as a biphasic disease; during the initial phase, patients exhibit influenza-like symptoms, which are subsequently followed by CNS or neurological manifestations in the later stage. The progression of TBE is influenced by factors such as the viral subtype, the age of the patient, and genetic predispositions [225,228,229]. Nevertheless, the mechanisms by which *TBEV* traverses the BBB remain largely unresolved. The researchers have demonstrated that the accumulation of *TBEV* in the brain occurs prior to the disruption of the BBB, suggesting that enhanced permeability of the BBB is not a necessary condition for *TBEV* infection within the brain [230]. Martin Palus et al. posited that *TBEV* could potentially penetrate the brain through a transcellular route without compromising BBB integrity [231]. Additionally, Zhou et al. proposed that *tick-borne flaviviruses* might invade host cells via exosomes released by arthropods, with exosomes originating from brain endothelial barrier cells facilitating the transfer of *tick-borne flavivirus* infectious RNA and proteins among neuronal cells [232]. Furthermore, Gould et al. have suggested that retroviruses may utilize the budding and transport mechanisms of exosomes to escape immune detection [233]. Consequently, the involvement of exosomes in *TBEV* infection represents a compelling area for further research and exploration.

#### 7.4.2. Exosomes and Neuroinflammation

Neuroinflammation is an immune response activated by microglia and astrocytes within the central nervous system, often implicated in the pathogenesis of various neurological disorders. It has been reported that upon neuroinflammatory events, such as stroke, there is an upregulation of miR-141-3p expression [234]. Research conducted by Manoshi Gayen and colleagues revealed that human astrocytes induced in an inflammatory milieu exhibit a significant increase in the levels of specific miRNAs, particularly miR-141-3p and miR-30d, in their released exosomes compared to their normal physiological resting state [235]. This suggests that exosome-derived miRNAs may serve as potential biomarkers for the diagnosis of neurological diseases. Che et al. utilized exosomes obtained from human umbilical cord mesenchymal stem cells (hUC-MSC), which markedly mitigated induced neuroinflammation and oxidative stress in vitro, while effectively facilitating the shift of microglia from a pro-inflammatory (M1) state to an anti-inflammatory (M2) state [236]. He and colleagues demonstrated that exosomes derived from astrocytes containing lncRNA 4933431K23Ri attenuate the pro-inflammatory phenotype of microglia, influencing their migration, chemotaxis, and phagocytic activity. This process alleviates neuroinflammation and enhances neuronal function, suggesting potential therapeutic targets for neuroinflammatory disorders [237]. Serving as an information conduit between cells, exosomes exhibit significant potential in modulating inflammation, suggesting their substantial promise as an anti-inflammatory agent for the treatment of cerebral pathologies.

#### 7.4.3. Exosomes and Epilepsy

Epilepsy is a paroxysmal neurological disorder characterized by disturbances in cerebral electrical activity that disrupt normal communication between neuronal cells, leading to seizure episodes. These episodes are manifested as recurrent muscle rigidity, convulsions, or loss of consciousness [238]. Currently, there are no effective pharmacological treatments to prevent the onset of epilepsy, with phenobarbital (PhB) being the most widely utilized antiepileptic drug. However, its efficacy fluctuates due to influences from genetic polymorphisms and hereditary variation. An increasing body of evidence suggests that exosomes may play a significant role in the treatment of epilepsy [238,239], Long et al. utilized A1-exosomes derived from human bone marrow mesenchymal stem cells administered intranasally to treat mice in a state of status epilepticus (SE) [240]. Their research demonstrated that A1-exosomes could reach the hippocampus within six hours, helping to suppress the elevation of pro-inflammatory cytokines while increasing anti-inflammatory factor concentrations. The neuroprotective effects mediated by A1-exosomes significantly reduced the loss of GABAergic interneurons within the hippocampus. Moreover, the intranasal delivery of A1-exosomes was found to maintain sustained anti-inflammatory effects and promote normal hippocampal neurogenesis during the chronic phase following SE. These findings indicate that exosomes have promising potential as drug delivery vehicles for epilepsy treatment (Table 3).

**Table 3 ijms-26-02491-t003:** The therapeutic potential and underlying mechanisms of exosomes and their cargo in the treatment of neurological disorders.

Disease	Origin of Exosomes	Exosome Content	In Vitro	In Vivo	Outcomes	Ref.
TBI	Astrocyte	miRNA-873a-5p	LPS-induced primary microglia	Male C57BL/10ScNJ mice	Inhibit the phosphorylation of ERK and the NF-κB signalling pathway	[149]
TBI	HucMSC	-	Mouse cortical neurons	Male ICR mice	↓ Neuron cell death, suppressed apoptosis, pyroptosis, and ferroptosis, ↑ the PINK1/Parkin pathway	[150]
TBI	BMSC	-	NSC	Male SD rat	Achieved endogenous NSC recruitment and neuronal differentiation, and promoted angiogenesis	[151]
TBI	hNSC	-	-	Male Wistar rat	Improved neurobehavioral performance after TBI	[152]
TBI	NSC		LPS-induced Microglial cell (BV2)	Male SD rat	Promoted NSC differentiation and reduced neuroinflammation	[153]
Stroke	NSC	-	-	Male C57BL/6 mice	Improved brain tissue damage such as cerebral infarction, neuronal death, and glial scar formation, and promote motor function recovery	[156]
Stroke	hNSC	BDNF	H_2_O_2_-induced NSC	Male SD rat	Inhibited the activation of microglia and promoted the differentiation of endogenous NSCs into neurons	[157]
Stroke	Astrocyte	miRNA-34c	The mouse neuroblastoma cell (N2a)	Male Wistar rat	Downregulation of NF-κB/MAPK axis and alleviation of I/R-induced nerve damage	[158]
Stroke	Panax notoginseng	-	Primary microglia	Male SD rat	Reduced I/R injury and improved behavioral outcomes	[160]
Stroke	ADSC	miRNA-760-3p	The mouse neuroblastoma cell (N2a)	Male C57/BL6 mice	Improved the neurobehavioral function of mice after I/R and inhibited ferroptosis	[161]
Stroke	ADSC	miRNA-19b-3p	-	C57BL/6 mice	Improved the neurological function of mice and inhibited ferroptosis	[162]
Stroke	ADSC	-	-	Male C57/BL6 mice	Inhibited M2 microglia ferroptosis and improved neurological function in ischemic stroke mice	[241]
AD	Dendritic cells	GAPDH siRNA	Neuro2A and SH-SY5Y	Male C57BL/6 mice	↓ The BACE1 mRNA ↓ Aβ 1–42	[172]
AD	Mesenchymal stem cell-derived	-	Microglia cells	Female triple-transgenic AD mice	↓ Neuroinflammation and tau pathology	[173]
AD	Serum	fasudil	Hippocampus tissues	Male APP/PS1 mice	The mmu-miR-451a and mmu-miR-19a-3p can enhance cognitive function	[174]
AD	HEK-293T	miR-29b	U87 cells	Male Wistar rats	↓ Amyloid-β (Aβ) peptide	[242]
AD	Mesenchymal stem cell-derived	-	FAD human neural cell	AD transgenic mice	↓ A β expression	[177]
PD	MSCs	ASO	SH-SY5Y cells	A53T α-syn transgenic mice	↓ The expression of α-syn and attenuated its aggregation	[187]
PD	hucMSCs	-	SH-SY5Y cells	Male SD rats	Promoting dopaminergic neuron survival in a Parkinson’s disease model, ↓ neuroinflammation, and improve the motor function	[189]
PD	UCB	-	MN9D cells and SH-SY5Y cells	Male C57BL/6 mice	Inhibition of hyperphosphorylation of MAPK p38 and ERK 1/2 signaling pathways	[190]
PD	HEK293T self-assembly	CA	Neuro-2a, iPC12, HeLa cell	C57BL/6 mice	Improve motor function, ↓ neuronal loss, ↓ α-syn pathological burden	[188]
PD	NSC	-	SH-SY5Y, BV2	Male C57BL/6 mice	↓ ROS and proinflammatory cytokines	[192]
PD	Umbilical cord blood mononuclear cell	MicroRNA-124-3p	N27 dopaminergic cells, NSC	Male C57BL/6	Protecting the dopaminergic neurons in the nigra and striatal fibers	[193]
PD	HEK-293T	Therapeutic catalase mRNA	Neuro2A cell, HEK-293T	Female C57BL/6 J mice	↓ Neurotoxicity and neuroinflammation	[194]
ALS	Schwann cells	-	-	An 81-year-old male patient	No adverse effects	[197]
ALS	ASCs from inguinal adipose tissues of C57Bl6/J mice	-	-	Transgenic mice overexpressing human SOD1 carrying a Gly93-Ala mutation and WT mice (B6SJL)	↓ The glial cells activation	[198]
ALS	hBMEPC	-	mBEC	-	↓ The damage of mBECs	[199]
glioma	MSCs	miR-199a	U251	Female Balb/c nude mice	↓ AGAP2	[207]
glioma	Raw264.7 cells	SPIONs, Cur, RGE	U251	Female BALB/c nude mice	Image, synergistic antitumor effect	[92]
glioma	Marrow stromal cells	miR-146b	9L gliosarcoma cells	Male Fischer rats	Anti-tumor effect	[208]
glioma	U87MG	Selumetinib	U87MG, A549	Male Balb/c-nude mice	Specific antitumor effect	[209]
GBM	Mouse brain endothelial bEnd.3 cells	NP_DOX_	glioma GL261 cells	C57BL/6 male mice	Induce apoptosis and ICD	[210]
glioma	NK cells	RSL3	C6 glioma cells, bEnd.3 cells	Both male and female ICR mice	↑ Ferroptosis and immune activation	[211]
Neuroinflammation	hUC-MSC	-	BV-2 cells, Primary microglia	Male C57BL/6J mice	Inhibit the microglial NRF2/NF-κB/NLRP3 signaling pathway	[236]
Status epilepticus	hMSCs from bone marrow	-	-	Male C57BL/6J mice	Neuroprotective and antiinflammatory effects	[240]

↑, Upregulation/promote; ↓, Downregulation/suppress; TBI, traumatic brain injury; LPS, lipopolysaccharide; ERK, extracellular-regulated protein kinases; NF-κB, nuclear factor-κB; ALS, amyotrophic lateral sclerosis; ASCs, adipose-derived stem cells; WT, wild-type; hBMEPC, human bone marrow-derived endothelial progenitor cells; mBEC, cells from mouse brain endothelial cell line bEnd.3; MSCs, mesenchymal stem cells; AGAP2, Arf GTPase-activating protein-2; SPIONs, iron oxide nanoparticles; Cur, curcumin; RGE, neuropilin-1-targeted peptide(RGERPPR); U87MG, human glioblastoma cells; GBM, glioblastoma; NP_DOX_, DOX-loaded nanoparticles; HucMSC, human umbilical cord mesenchymal stem cell; PINK1/Parkin, PTEN induces the presumed protein kinase 1 and the E3 ubiquitin ligase Parkin; BMSC, bone marrow mesenchymal stem cell; NSC, neural stem cell; hNSC, human neural stem cell; MAPK, mitogen-activated protein kinase; I/R, ischemia/reperfusion; ADSC, adipose-derived mesenchymal stem cell.

## 8. Clinical Research on Brain Diseases

Due to their unique biological characteristics, including the ability to cross the blood–brain barrier, low immunogenicity, and good biocompatibility, exosomes have become a hot research area for the diagnosis and treatment of brain diseases. Currently, several clinical trials are exploring the application value of exosomes in Alzheimer’s disease, Parkinson’s disease, stroke, and other neurological diseases. Researchers are mainly conducting research in two directions: on the one hand, utilizing specific biomolecules carried by exosomes as biomarkers for early diagnosis and prognosis assessment of diseases; on the other hand, exosomes are used as an important component of novel cell-free therapeutic strategies [243].

Of the five clinical trials listed on ClinicalTrials.gov evaluating exosomes as biomarkers for brain disorders, two have concluded their respective studies. Xintong Ge et al. are investigating potential biomarkers of RNA and protein cargo in the blood and exosomes of TBI patients [244]. Furthermore, Carlo Walter Cereda et al. are examining EVs as biomarkers in patients with transient ischemic attack (TIA) to differentiate between the ischemic process and stroke mimics [245]. Researchers from the Tang-Du Hospital are exploring exosomes as early serum markers and potential intervention targets for disease monitoring in patients with cerebral hemorrhage [246]. Researchers from Fondazione Don Carlo Gnocchi Onlus have completed a clinical study assessing the prognostic capacity of circulating Evs in the serum of stroke patients [247]. Additionally, researchers from the University of Alabama at Birmingham have concluded a clinical trial in Parkinson’s disease patients exploring exosome biomarkers associated with disease progression, though the results are pending [248]. Despite the promising therapeutic potential of exosomes, no clinical trials for exosome-based treatments for brain disorders have been completed to date. Researchers from the China Medical University Hospital are investigating the role of acupuncture-induced exosomes in treating post-stroke dementia, with participant recruitment ongoing [249]. Leila Dehghani, et al. have analyzed the efficacy of MSC-derived exosomes in patients with acute ischemic stroke [250]. Furthermore, Junwei Hao et al. are evaluating the safety and efficacy of the intravenous administration of exosomes derived from human induced pluripotent stem cells (GD-iExo-003) in acute ischemic stroke patients [251]. In a separate study, they formulated exosomes derived from human umbilical cord mesenchymal stem cells (hUC-MSC-sEV-001) into a nasal drop for intranasal delivery in the treatment of ALS [252]. Researchers from the Ruijin Hospital have initiated a Phase I/II clinical trial to explore the efficacy of exosomes derived from allogeneic adipose-derived mesenchymal stem cells (MSCs-Exos) for Alzheimer’s disease [253] (Table 4).

Currently, a substantial body of preclinical and clinical research has substantiated the therapeutic potential of exosomes across a spectrum of diseases, including neurological disorders and conditions affecting the digestive system, ocular tissues, and cardiac tumors [254]. Furthermore, a growing number of clinical trials involving exosomes are concentrated on two primary disease categories: neurological disorders (67 trials, 15%) and cancer (194 trials, 43%). Despite the successful modification of exosome cargo to facilitate the delivery of anticancer nucleic acids and drugs, as well as the application of exosome membranes targeting specific tumor sites, thereby surmounting several technical hurdles, the precise interactions between exosomes and tumor cells remain a subject of ongoing investigation [255,256,257,258]. Clinical trials employing exosomes for the treatment of neurological disorders are predominantly in the early phases (Phase I/II). While preliminary clinical data on exosomes appear promising, potential challenges persist, such as standardization of production, dosage determination, and heterogeneity, which may influence subsequent trial outcomes [259]. Consequently, the standardization of exosome dosage and scalable manufacturing processes remain significant bottlenecks in the clinical translation and industrialization of exosomes [260]. Moreover, the regulatory landscape surrounding exosomes is complex, owing to their unique intracellular mechanisms of action, and varies across different countries. The diversity of production technologies further complicates standardization efforts [261].

In summary, this review systematically synthesizes strategies for utilizing exosomes as brain delivery vehicles and their advancements in the treatment of brain disorders. The core content focuses on a comprehensive overview of the current status of exosome-based clinical treatments for brain diseases, aiming to provide insights for future industrial development and exploration of exosomes.

**Table 4 ijms-26-02491-t004:** Clinical trials of exosomes in the treatment of neurological disorders.

Therapeutic Applications	Exosome Subtypes	ClinicalTrials.gov ID	Ref.
TBI	Blood-derived exosomes	NCT04928534	[244]
Intracerebral Hemorrhage	Circulating exosomes	NCT05035134	[246]
Post-Stroke Dementia	-	NCT05326724	[249]
Stroke	Serum exosomes	NCT05370105	[247]
Stroke	Mesenchymal stem cell	NCT03384433	[250]
Acute Ischemic Stroke	Human-induced pluripotent stem cell (GD-iExo-003)	NCT06138210	[251]
Stroke	Blood-derived exosomes	NCT06319742	[245]
AD	Allogenic adipose mesenchymal stem cells (MSCs-Exos)	NCT04388982	[253]
PD	-	NCT01860118	[248]
ALS	Human umbilical cord blood mesenchymal stem cells (hUC-MSC-sEV-001)	NCT06598202	[252]

## 9. Conclusions and Future Perspectives

So far, exosomes have increasingly been recognized as a promising candidate for an innovative drug delivery nanoplatform geared towards the diagnosis and treatment of neurological disorders. Their distinctive biologically derived membrane architecture promotes the transport of nucleic acids and therapeutic agents. Additionally, their inherent high biocompatibility, low toxicity, and minimal immunogenicity underscore their substantial potential as therapeutic instruments. Exosomes demonstrate enhanced stability in circulation, which prolongs drug pharmacokinetics, consequently boosting their bioavailability and therapeutic efficacy. By engineering these exosomes, we can integrate specific targeting functionalities that facilitate precise recognition and binding to intended cells or tissues, significantly improving therapeutic outcomes while reducing side effects. Furthermore, exosomes typically exhibit unique permeation capabilities, allowing them to cross the blood–brain barrier effectively. In comparison to cellular therapies, the preparation of exosomes is relatively straightforward and can be refined by modifying the cellular environment to tailor their functionalities. Additionally, exosomes are less likely to elicit immune rejection due to their non-proliferative nature, thereby mitigating the risks associated with cellular therapies, including vascular occlusion and tumorigenesis. However, while exosomes exhibit unique advantages in brain delivery, their efficiency in traversing the BBB is influenced not only by disease induction and modulation but also by enzymatic activity, clearance kinetics, and microcirculation dynamics. Despite advancements in carrier modification for targeted BBB penetration, limited drug permeation persists, as the binding of targeting ligands to drug carriers does not enhance their brain biodistribution; instead, it primarily improves cellular internalization upon encountering target cells. Furthermore, several limitations and challenges must be addressed before exosomes can be widely adopted in clinical practice. For instance, exosomes exhibit significant heterogeneity in terms of their origin, structure, function, and cargo. The potential for immunogenicity may also elevate therapeutic risks, particularly in patients with compromised immune systems. The standardization of exosome isolation and preparation processes remains incomplete, and the absence of unified isolation standards could compromise the safety and efficacy of exosome-based therapies. Therefore, a more balanced approach to the therapeutic application of exosomes, aimed at enhancing treatment outcomes and safety, necessitates intensified basic research, the establishment of standardized isolation and preparation protocols, the assessment of immunogenicity risks, improved inter-batch quality control, and the conduct of clinical trials with long-term follow-up. Moreover, the development of a unified quality standard for exosome-related regulatory oversight would facilitate the industrialization of exosomes.

In summary, a more comprehensive understanding of how exosomes traverse the BBB, along with their mechanisms of action and resultant effects, will be instrumental in the design and development of novel therapeutic strategies for neurological disorders. It is also critical to acknowledge the impact of microcirculatory dysfunction on the treatment of brain diseases. Therefore, future research should focus on designing and optimizing drug-encapsulating exosomes to maintain or restore microvascular vasodilation, selectively augment blood flow to the affected brain regions, thereby enhancing drug accumulation within the lesion. Furthermore, the ability to selectively “turn on” and “turn off” drug permeation pathways in diseased or healthy brain regions in a real-time, controlled manner is of paramount importance. Beyond these considerations, the standardization of exosome clinical applications is essential to explore and implement innovative, exosome-centered therapeutic strategies for brain disorders.

## Figures and Tables

**Figure 1 ijms-26-02491-f001:**
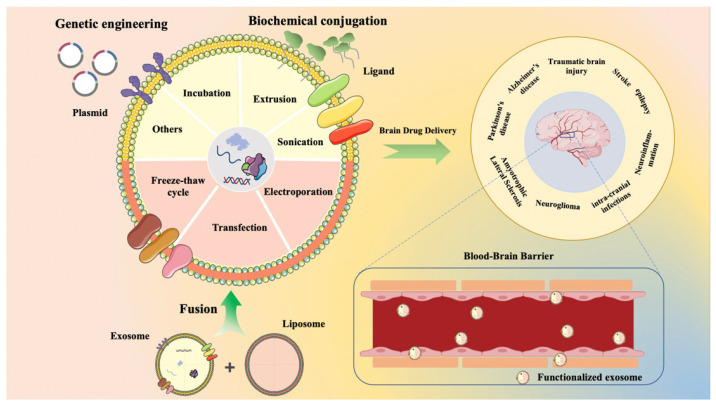
A diagram illustrating the isolation, functionalization, and drug loading of exosomes, as well as the penetration of drug-loaded exosomes across the blood–brain barrier for the treatment of brain disorders.

**Figure 2 ijms-26-02491-f002:**
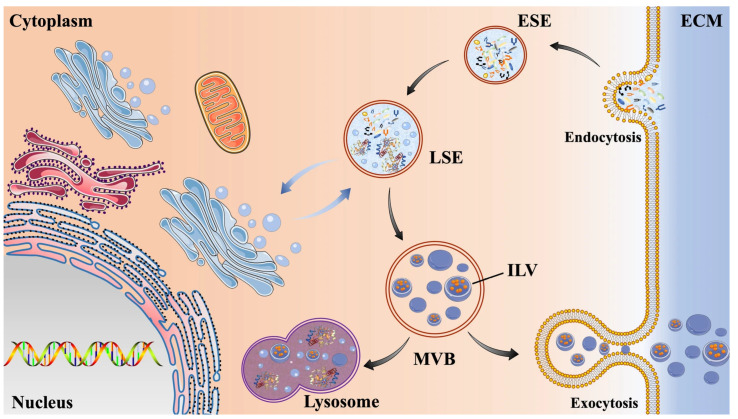
The secretion process of exosomes. Extracellular components can enter cells through endocytosis and plasma membrane depression. The vesicles formed during this process can be fused with the early sorting endosomes (ESEs), which are then transformed into the late sorting endosomes (LSEs). The second invagination in LSEs leads to the production of intraluminal vesicles (ILV). LSEs are further transformed into multivesicular bodies (MVBs), which can be degraded by fusion with lysosomes or autophagosomes, or they can be fused with plasma membrane to release ILVs as exosomes. ECM, extracellular matrix [39]. Copyright 2022, the author(s).

**Figure 3 ijms-26-02491-f003:**
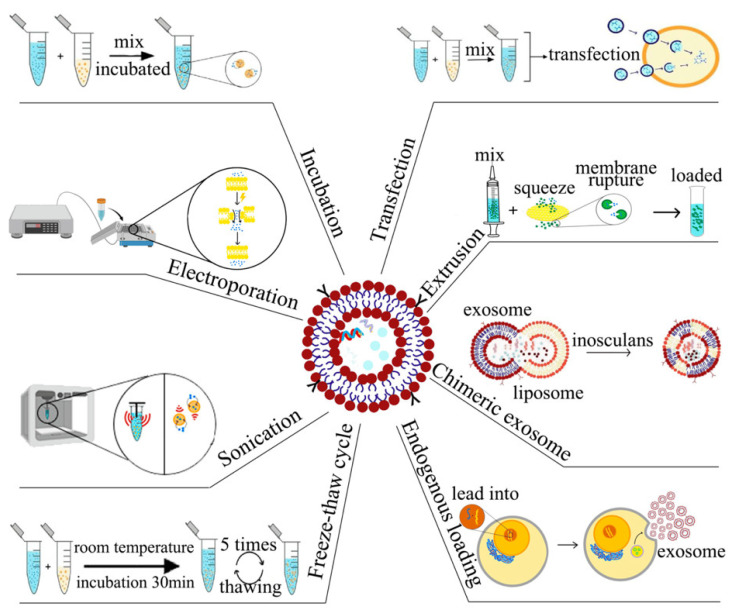
Schematic representation of the exosome-based drug loading methodology [104]. Copyright 2023 by the authors.

**Figure 4 ijms-26-02491-f004:**
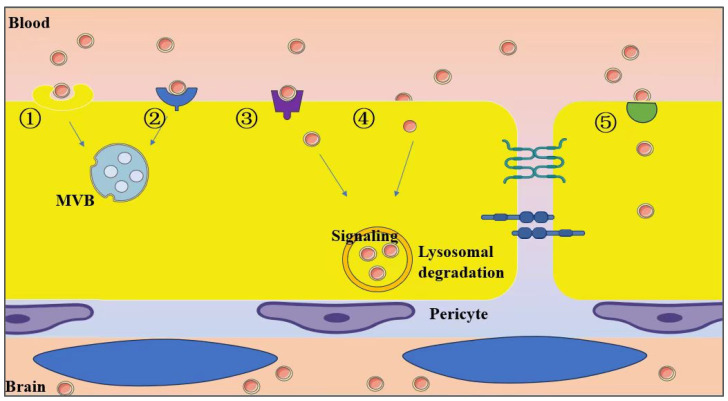
A schematic representation of five hypothesized pathways for the action of exosomes at the blood–brain barrier via brain endothelial cells. ① Exosomes are incorporated into cells via reverse membrane invagination. ② Exosomes enter cells via receptor-mediated endocytosis. ③ Non-specific lipid rafts facilitate exosome transport. ④ Exosomes adhere to and fuse with the endothelial cell membrane, thereby entering the cell. ⑤ Exosomes bind to G protein-coupled receptors (GPCRs) on the cell surface to induce signaling cascades. Three outcomes are anticipated following transcytosis: signal induction, lysosomal degradation, or transport through brain endothelial cells.

**Figure 5 ijms-26-02491-f005:**
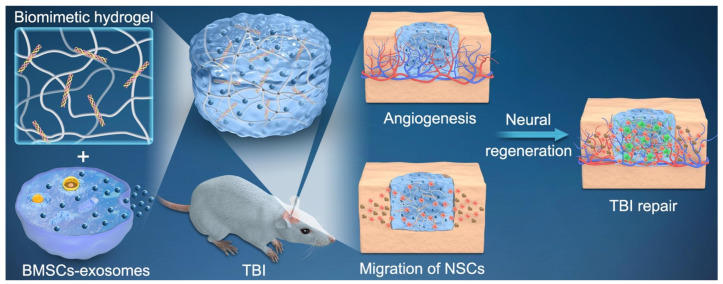
The synergistic effects of bone marrow mesenchymal stem cell-derived exosomes incorporated into hyaluronic acid-collagen hydrogels (DHC-BME) on the recruitment and differentiation of endogenous NSCs, along with angiogenesis, promote neurological functional recovery following TBI in rats [151]. Copyright 2023 Elsevier Ltd. All rights reserved.

**Figure 6 ijms-26-02491-f006:**
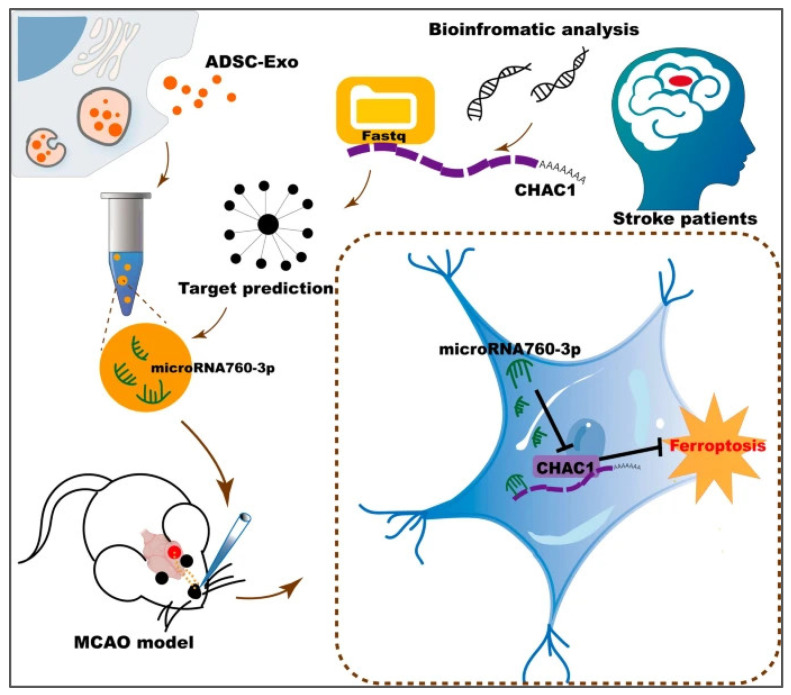
Exosomes (ADSC-Exo) derived from adipose-derived mesenchymal stem cells, and containing miR-760-3p, ameliorate ferroptosis via targeting CHAC1 for stroke therapy [161]. Copyright the author(s), 2023.

**Figure 7 ijms-26-02491-f007:**
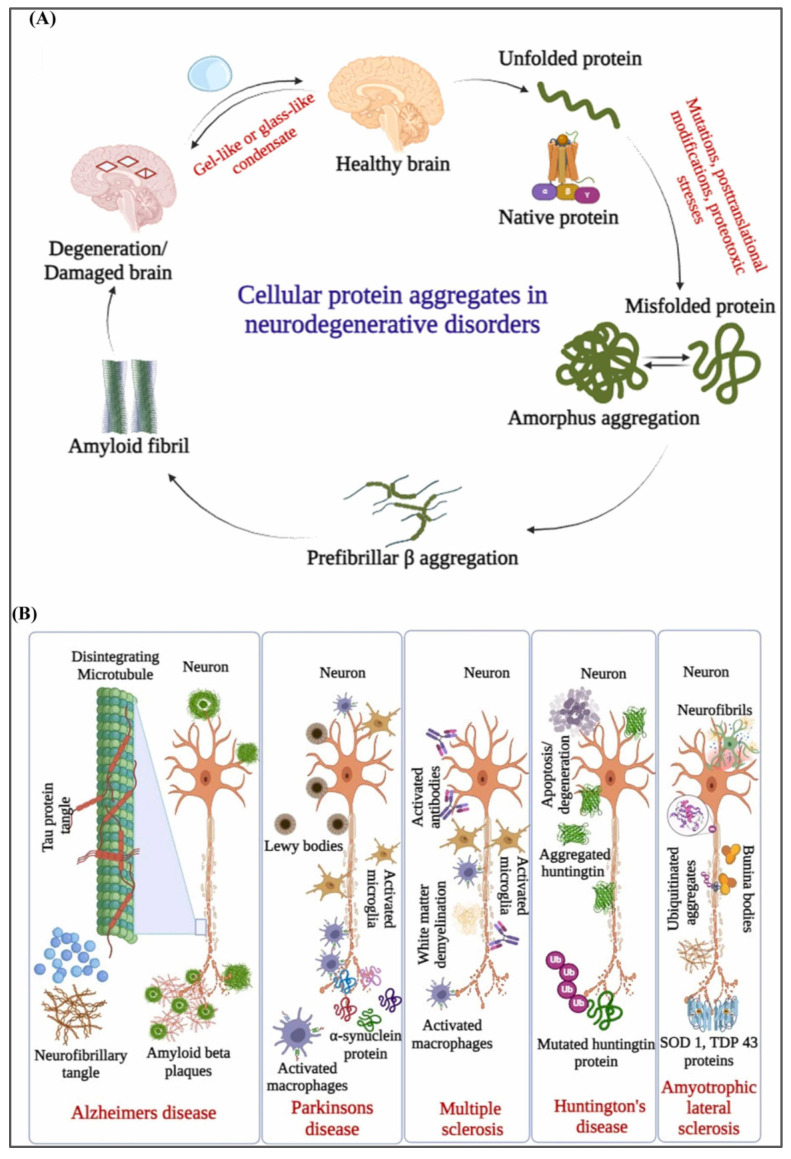
Schematics of the steps involved in cellular protein aggregation associated with neurodegeneration: (**A**) represents the native structure of protein transformed in misfolding due to stresses, which impact amyloid protein fibrils and initiation of neurodegeneration. (**B**) Different proteins and their aggregates are involved in AD, PD, MS, HD, and ALS [163]. Copyright 2024, the author(s).

**Figure 8 ijms-26-02491-f008:**
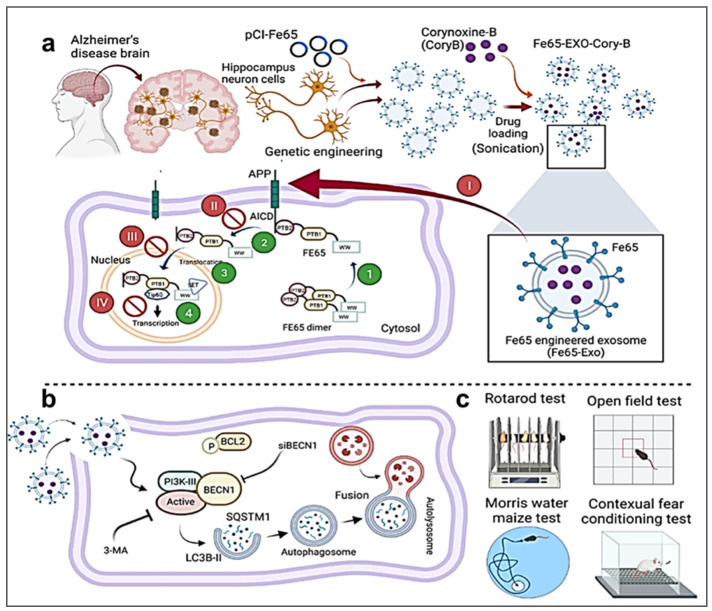
A new targeted drug delivery system using engineered exosomes (Fe65-EXO) from hippocampal neurons is introduced. These exosomes deliver Cory-B to neurons, promoting autophagy, which enhances cognitive function and reduces AD pathology in mice. (**a**) Representative schematic diagram showing the APP-targeted delivery of Cory-B loaded in Fe65 engineered hippocampus neuron cells-derived exosomes. (**b**) Potential role of Fe65-EXO encapsulating Cory-B (Fe65-EXO-Cory-B) in the induction of autophagy in neuron cells via BECN1 or ATG5, or ATG7, leading to (**c**) the improvement of cognitive and locomotor behavior for ameliorating pathogenesis in murine AD mouse, as detected via rotarod, open field, Morris water maize, and contextual fear conditioning test [90]. Copyright 2023, the author(s).

**Figure 9 ijms-26-02491-f009:**
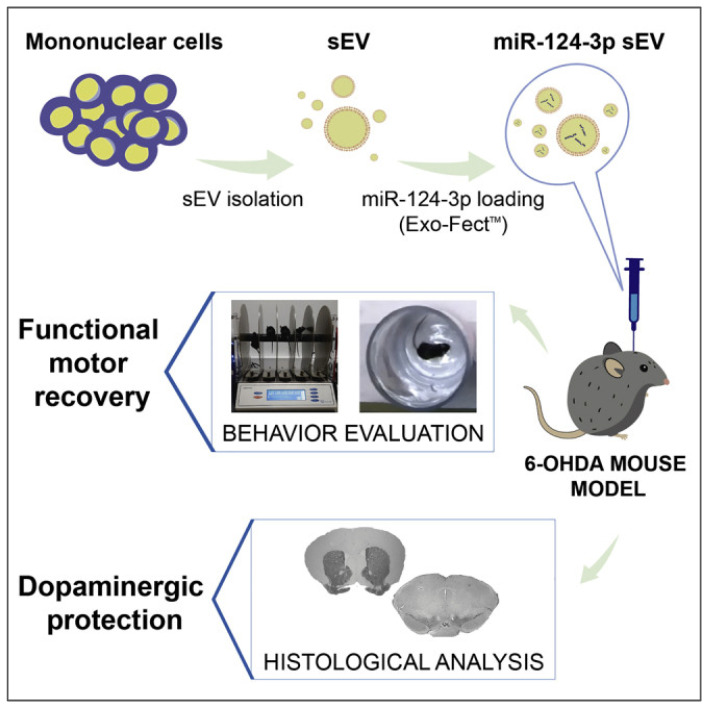
In the context of Parkinson’s disease, umbilical cord blood mononuclear cell-derived small extracellular vesicles, utilized as a miR-124-3p delivery vehicle, demonstrate the capacity to safeguard dopaminergic neurons within the substantia nigra and striatal fibers, concurrently facilitating the restoration of motor function in PD murine models [193]. Copyright 2022, the authors.

**Figure 10 ijms-26-02491-f010:**
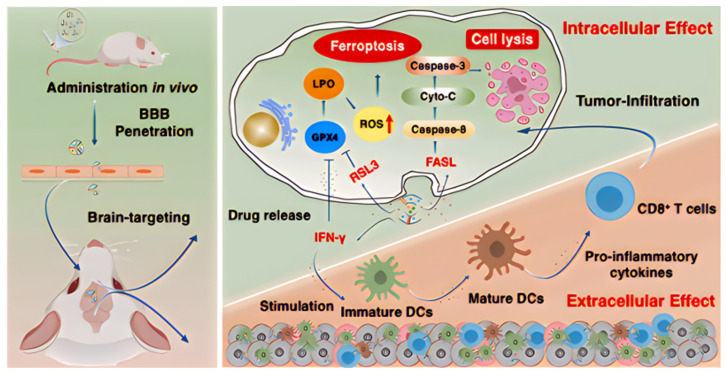
hNRVs mediate a positive feedback loop between ferroptosis therapy and immunotherapy for the treatment of gliomas. hNRVs exhibit tumor-homing effects, leading to their active accumulation and enhanced cellular uptake within the tumor microenvironment. The release of FASL, IFN-γ, and RSL3 into the tumor microenvironment results in the effective cleavage of tumor cells by FASL. RSL3 downregulates the expression of GPX4 in tumors, leading to the accumulation of LPO and ROS, and promoting ferroptosis in tumor cells. The accumulation of IFN-γ and TNF-α stimulates the maturation of dendritic cells, effectively inducing GPX4 inactivation, promoting lipid peroxidation, and indirectly promoting the occurrence of ferroptosis [211]. Copyright 2024, American Chemical Society.

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
