# Peer review of "Harnessing the Potential of Exosomes in Therapeutic Interventions for Brain Disorders"

_ijms, 2025, doi:10.3390/ijms26062491_

Round 1
Reviewer 1 Report (New Reviewer)
Comments and Suggestions for Authors
Dear Authors;
Re: [Manuscript ID; ijms-3501512]
Title: "Harnessing the Potential of Exosomes in the Therapeutic Intervention of Brain Disorders"
In this manuscript you aimed to address the opportunities and challenges associated with utilizing exosomes as a drug delivery system for the brain, summarizing the barriers to clinical translation and proposing future research directions.
Please find my comments below:
- There is no gap between the last word in the sentence and [References] throughout the manuscript. This is a serious punctuation error and is present in your article in numerous occasions.
- There is no Reference at all for the text in Lines 80-102.
- Why Figure 1 is designated as GRAPHICAL ABSTRACT?
- There is hardly any original Scheme or Figure or Illustration in your article.
- In the Figure Legends the expression "The Authors" is typed with Capital Letters and in other occasions with Small Letters "the authors'. Please be consistent.
- In Line 155, why the beginning of sentence started with "small letter"? (i.e., ... approaches. outline their ...).
- The end of the sentence in Line 173 has two fullstops (i.e., ... its biological activity.[52]. ...).
- Avoid starting a sentence with "And" (e.g., see Line 289).
- In Line 331 and 332 you repeated the definition of exosomes for several times.
- Section 5 has no References at all (Lines 331-340).
- In Line 404 the sentence started with small letter (... efficiency[108]. the extrusion technique...).
- The text in Lines 404-413 needs References.
- Please check the sentence in Line 860: "... as a site for immunological activation (Error! Reference source not found.), offering ...".
- The text in Lines 857-872 needs References.
- Please check the expression in Lines 903-904: "... (Error! Reference source not found.)"!
- Please highlight/ make clear: What is the significance of arrows in Table 3?
- In the Conclusions, you mentioned: "... Furthermore, the ability to selectively "turn on" and "turn off" drug permeation pathways in diseased or healthy brain regions ...". This needs to be more clearly described in the previous sections of the manuscript.
Author Response
Please see the attachment.

Reviewer 2 Report (New Reviewer)
Comments and Suggestions for Authors
The manuscript offers a critical review of exosomes for use as vehicles for treating brain disorders. The review, though very applicable, has not been conducted with the necessary methodological stringency and critical examination of the research. The review needs significant revisions in the following areas:
The review has no apparent methodology. There's nothing mentioned of systematic searching of the literature, inclusion and exclusion, and critical appraisal of the studies selected. The authors must present a systematic process of selecting the literature for purposes of strengthening the validity of the review.
While the paper extensively discusses exosomes, it doesn't compare them with other nanoparticle-based systems of drug delivery critically. A critical comparison with other alternative carriers (e.g., polymeric nanoparticles, liposomes, viral vectors) would situate the advantages and limitations of exosome-mediated delivery.
The manuscript contains too technical detail on exosome biogenesis and methods of loading and not enough on their clinical application. The authors must invest more effort into translating such research into practice, addressing barriers of regulations, scalability, and use in real settings.
While the review does touch on difficulties, it doesn't critically discuss the major shortcomings of exosome treatment, such as heterogeneity, standardisation of isolation, potential for being immunogenic, and batch-to-batch heterogeneity. A more balanced approach needs to be made not to overstate the therapeutic value of exosomes.
The manuscript contains several figures, and they are not very seamlessly integrated into the text. Some of the figures are not properly captioned and explained, and hence are not easy to understand. The authors must ensure that figures are synchronised with the discussion and reinforce rather than hinder the flow of data.
Overall, the paper has valuable insights, but substantial reorganisation and critical sophistication are necessary for purposes of a high-impact review. Resolution of these matters will strengthen and amplify the pragmatic value of the research.
Major revision is advised. Please highlight all revision in yellow.
Round 2
Reviewer 2 Report (New Reviewer)
Comments and Suggestions for Authors
The authors have made substantial revisions, addressing key concerns regarding methodology, comparison with other delivery systems, and clinical applicability. While they have improved the structure and balance of the review, the methodological approach remains somewhat descriptive rather than rigorously systematic. Overall, the manuscript is significantly strengthened.
This manuscript is a resubmission of an earlier submission. The following is a list of the peer review reports and author responses from that submission.
Round 1
Reviewer 1 Report
Comments and Suggestions for Authors
This manuscript reviews the usefulness of exosomes in the therapy for brain disorders. In a whole of manuscript, the text is too long to understand the concepts and contents. This reviewer cannot understand the main point of this review. In addition, this manuscript has the many original data without materials & methods and peer-review. In few example review, the review includes one or two original figures to reinforce the purpose of the review, however, this manuscript has too many figures. This is not acceptable for the publication. This reviewer strongly recommends that the authors should review the issues and fundamentally reconsider the main focus and structures of the review. Also, this reviewer strongly recommends that the authors should remove the original data from the manuscript. There would be little benefit to the readers.
Comments on the Quality of English LanguagePlz see the comments.
Reviewer 2 Report
Comments and Suggestions for Authors
The manuscript titled “Harnessing the Potential of Exosomes in the Therapeutic Intervention of Brain Disorders” by Bai, L.; et al. is a Review work where the authors outlined the most recent advances in the use of exosomes as drug delivery for brain malignancies. Exosomes overcome some limitations offered by other nanomaterials as magnetic nanoparticles. For this reason, exosome-based therapies have emerged as promising approach to treat patients with brain disorders. The manuscript is generally well-written and this is a topic of growing interest.
However, it exists some points that need to be addressed (please, see them below detailed point-by-point) to improve the scientific quality of the submitted manuscript paper before this article will be consider for its publication in the International Journal of Molecular Sciences.
1) Keywords. The authors should consider to add the term “biocompatibility” in the keyword list.
2) “In recent decades, neurodegenerative disorders (…) significant burden on society” (lines 26-27). Could the authors provide quantitative data insights according to the worldwide global burdens of neurodegenerative diseases and their economic impact cost on society? This will significantly aid the potential readers to better understand the significance of the topic covered in this Review work.
3) Then, it would be also neccesary to provide some details according the onset, proliferation and progression of neurodegenerative disorders with some schematic representation (eventually some information from the sections 7.2.1.; 7.2.2. and 7.2.3 could be moved here). In this context, the effect of the existing divalent ions [1] or the ionic strength [2] inner the brain tissue are pivotal factors to consider.
[1] Carapeto, A.P.; et al. Morphological and Biophysical Study of S100A9 Protein Fibrils by Atomic Force Microscopy Imaging and Nanomechanical Analysis. Biomolecules 2024, 14, 1091. https://doi.org/10.3390/biom14091091
[2] Ziaunys M.; et al. Polymorphism of Alpha-Synuclein Amyloid Fibrils Depends on Ionic Strength and Protein Concentration. Int. J. Mol. Sci. 2021, 22, 12382. https://doi.org/10.3390/ijms222212382
4) “In the context of biogenesis the endosomal sorting complex required for transport (…) membrane invagination and promotes membrane modifications” (lines 115-117). It should not be neglected the role of ESCRT protein family on the membrane reparation processes inner the cell.
5) “These steps are designed to efficiently eliminate (…) (Error! Reference source not found)” (lines 151-153). Please, the authors should fix this issue and link the respective reference citation by the bibliography manager software tool.
6) “3.4. Size Exclusion Chromatography” (lines 212-233), “3.5. Polymer precipitation” (lines 233-253) and “3.6. Immunoaffinity” (lines 253-265). A schematic representation similarly as the Fig. 3 for the ultracentrifugation-based methods would be benefit the manuscript quality.
8) “8. Conclusions and future perspectives” (lines1274-1310). This section perfectly remarks the most relevant outcomes found by the authors in this field and also the promising future prospectives. It may be advisable to add a brief statement to remark the potential future action lines to pursue the topic covered in this topic.
Round 2
Reviewer 1 Report
Comments and Suggestions for Authors
The revised manuscript is improved but is still riddled with some serious issues. Except in special circumstances, it is not advisable to let the readers read the data in a review. The authors should have all the data and made it into a proper summary figure rather than just removing some of the data. This reviewer think that the authors fail to understand the essential problem of my concerns. Additionally, there is much overlap with what has been written in previous reviews about general exosomes, and the main points are still remains unclear.
Comments on the Quality of English Language
Plz see the comments.
